



# Error induced by neglecting subgrid chemical segregation due to inefficient turbulent mixing in regional chemical-transport models in urban environments

Cathy W. Y. Li[1], Guy P. Brasseur[1,2], Hauke Schmidt[1], and Juan Pedro Mellado[3]

[1]Max Planck Institute for Meteorology, Bundesstrasse 53, 20146, Hamburg, Germany
[2]National Center for Atmospheric Research, 1850 Table Mesa Dr, Boulder, CO 80305, USA
[3]Department of Physics, Aerospace Engineering Division, Universitat Politècnica de Catalunya, C. Jordi Girona 1-3, 08034, Barcelona, Spain

**Correspondence:** Cathy W. Y. Li (cathy.li@mpimet.mpg.de)

**Abstract.**

We employed direct numerical simulations to estimate the error on chemical calculation in simulations with regional chemical-transport models induced by neglecting subgrid chemical segregation due to inefficient turbulent mixing in an urban boundary layer with strong and heterogeneously-distributed surface emissions. In simulations of initially-segregated reactive

species with an entrainment-emission configuration with an A-B-C second-order chemical scheme, urban surface emission fluxes of the homogeneously-emitted Tracer A result in a very large segregation between the tracers and hence a very large overestimation of the effective chemical reaction rate in a complete-mixing model. This large effect can be indicated by a large Damköhler number ($Da$) of the limiting reactant. With heterogeneous surface emissions of the two reactants, the resultant normalised boundary layer-averaged effective chemical reaction rate is found to be in a Gaussian function of $Da$, and

is increasingly overestimated by the imposed rate with an increased horizontal scale of emission heterogeneity. Coarse-grid models with resolutions commensurable to regional models give reduced yet still significant errors for all simulations with homogeneous emissions. Such model improvement is more sensitive to the increased vertical resolution. However, such improvement cannot be seen for simulations with heterogeneous emissions when the horizontal resolution of the model cannot resolve emission heterogeneity. This work highlights particular conditions in which the ability to resolve chemical segregation

is especially important when modelling urban environments.

## 1 Introduction

Turbulence mixes initially-segregated reactive species in the boundary layer, and allows chemical reactions to occur. However, for fast chemical reactions with the chemical timescale shorter than the turbulent timescale, turbulent motions mix the reactants

so slowly that they remain segregated rather than reacting. In an Eulerian chemical-transport model, this chemical segregation,



when occurring at a length scale smaller than the size of a model grid box, cannot be represented as the chemical species are assumed to be instantaneously and homogeneously mixed within a grid box. The negligence of such subgrid segregation induces potential errors to the calculation of the chemical transformation within a model grid of a large-scale model.

Efforts have been made to examine and quantify this error under different turbulent and chemical regimes in a range of atmospheric environments. Earliest studies can be dated back to Damköhler (1940) and Danckwerts (1952), which focused on quantifying the effect of turbulence on combustion processes, also applicable to other chemical transformations such as chemical reactions, by introducing quantitative definitions of scaling parameters including the Damköhler number ($Da$, the ratio between the turbulent and chemical timescales) and the segregation coefficient ($I_S$ , the ratio between the covariance and the product of the mean concentrations). Donaldson and Hilst (1972) adopted the discussion in the context of atmospheric reactions. This line of research was then continued by a number of investigations that used first-order and second-order closure methods to study the profiles and budgets of the fluxes of chemical reactive species in the boundary layer. For example, Fitzjarrald and Lenschow (1983) and Vilà-Guerau de Arellano and Duynkerke (1993) reported that the chemical terms can be of similar importance as the dynamical term in the flux equations for the $NO-NO_2-O_3$ chemical system in the surface layer. Verver et al. (1997) further stated that ignoring the higher-order chemical terms in the flux equations would result in deterioration of model performance. As these closure methods failed to resolve vertical turbulent mixing and horizontal fluctuations, the investigation of the topic was extended with the use of large-eddy simulations (LES), which can resolve the most energetic eddies in the boundary layer.

Many of these LES studies focus on the convective boundary layer, in which the imbalance between updraft and downdraft transport produces a large segregation of the reactants (Wyngaard and Brost, 1984; Chatfield and A. Brost, 1987). Such LES studies were often performed with idealised cases with a bottom-up tracer emitted from the surface and top-down tracer entrained from the free troposphere with a simple second-order chemistry scheme. For instance, Schumann (1989) pointed out that the segregation between the two tracers depends on the Damköhler number ($Da$), the concentration ratio of the two species and the specified initial conditions, pinning the use of $Da$ as an indicator to estimate whether turbulent motions are significantly affecting chemical reactions in the flow. Vinuesa and Vilà-Guerau de Arellano (2005) introduced the concept of an effective chemical reaction rate ($k_{eff}$) to quantify the actual boundary layer-averaged reaction rate that accounts for the effect of the chemical segregation. They also reported a drop of $k_{eff}$ up to 20% from the imposed chemical reaction rate $k$ when $Da \sim 1$, while $k_{eff} \sim k$ when $Da \sim 0.1$. Recently, the effect of segregation due to inefficient turbulent mixing on chemical reaction has been considered as the cause of the miscalculation of large-scale models in a number of scenarios. One of the most discussed issues is the under-prediction of the concentration of hydroxyl radical (OH) in global models. A number of studies employed LES models over forestal areas with sophisticated chemical mechanisms involving biogenic VOCs to simulate the resultant segregation between isoprene and OH (e. g. Brosse et al. (2017); Dlugi et al. (2019)). However, the magnitude of the segregation coefficient $I_S$ shown in these studies is in general less than 20%, which is too small to explain the observed discrepancy, where $I_S \sim -50\%$ is necessary (Butler et al., 2008).

There are a number of factors that may increase chemical segregation. One possible factor is to increase the surface emission of the bottom-up tracer. For instance, Kim et al. (2016) indicated the segregation between isoprene and OH increases when



switching from very low- to very high-$NO_X$ surface emissions. Another possible factor is related to the emission heterogeneity of the bottom-up tracer. Krol et al. (2000) found that, by imposing non-homogeneous emission of a generic hydrocarbon (RH) at the surface, the segregation coefficient between RH and OH can reach up to -30%, compared to only -3% when RH is emitted homogeneously. Ouwersloot et al. (2011) implemented in their LES simulations patches of surface isoprene emissions with

different fluxes on the African savannah, and showed that the segregation increases with the width of the patches. Kaser et al. (2015) further pointed out that with the consideration of surface heterogeneity, chemical segregation can locally slow down the isoprene chemistry by up to 30%. Most of the previous LES studies focus on scenarios under rural conditions, but these two factors should be even more relevant for urban environments with intense emission of pollutants and high spatial heterogeneity.

The aim of the present work is to investigate the effect of inefficient turbulent mixing on chemical reactions in an urban-

like boundary layer with strong and heterogeneously-distributed surface emissions, and to account for the errors induced by neglecting the resultant subgrid chemical segregation in relatively coarse regional models. While previous studies focused on agricultural and rural conditions where the emission fluxes are relatively low ($\sim O$ (0.01) ppb m s$^{-1}$), our simulations address cases of strong emission fluxes in typical urban values ($\sim O$ (0.1-1.0) ppb m s$^{-1}$). In a rare related study in urban air condition, Auger and Legras (2007) obtained high values of instantaneous segregation under certain emission configurations

in urban areas with a chemical system of 44 species. On top of their conclusions, our study aims to explain why this strong segregation occurs under urban conditions and on which parameters the errors induced by neglecting the segregation in large-scale models depend. To achieve this aim, we perform DNS simulations with homogeneous emissions with an idealised second-order A-B-C chemistry scheme ($A + B \rightarrow C$) with emission fluxes extended to urban values, in addition to a set of simulations with heterogeneous emissions. With varied reaction rates, the idealised second-order A-B-C chemistry scheme can generally

represent any second-order chemical reactions commonly seen in an urban environment.

The results from our DNS simulations are then degraded into lower resolution to mimic the calculations from regional models. Previous studies often compared their LES results with a mixed-layer or complete-mixing model, which assumes the whole simulation domain to be in the same model grid. This assumption may still be reasonable for forested areas, as the global and mesoscale models in use typically have a mesh size of the order of 10 km, which is comparable to the size of

our simulation domain. However, this comparison is not suitable in application to urban environments, as the regional models simulating urban air quality typically have a much finer resolution with mesh size of the order of 1 km. Therefore, we degrade our DNS results to coarse-grid models with mesh size of 1 km and 3 km with multiple vertical resolutions to estimate the errors from regional models.

The structure of this paper is as follows. The next section introduces the DNS model adopted in this work and the settings

of the simulations. Subsequently, the results of our DNS simulations are first presented with cases of homogeneous emissions and then of heterogeneous emissions. This is followed by the comparison between the results from the degraded coarse-grid models and those from the DNS model to account for the errors from regional models. The implication of our results for regional models applied to urban environments is then discussed, and conclusions are provided at the end.



## 2  Model description

### 2.1  Dynamical settings

The relation between turbulence and chemistry in a convective boundary layer is investigated in this work by means of direct numerical simulation. There are two common approaches to numerically simulate turbulent flows, namely, large-eddy simula-
tion (LES) and direct numerical simulation (DNS). The former applies a low-pass filter and models the subgrid-scale effect on the filtered variables. The later solves the original Navier-Stokes equations but often with a molecular viscosity that is larger than the value in the real application represented (Monin and Mahesh, 1998; Pope, 2004; Mellado et al., 2018). Hence, both approaches are restricted to low to moderate Reynolds numbers. The Reynolds number is defined in terms of the subgrid-scale viscosity and numerical diffusivity in LES, and in terms of the molecular viscosity in DNS. It can be interpreted as the scale
separation between the largest and the smallest resolved scales (i. e. the Kolmogorov and Batchelor scales in DNS), which is in turn related to the number of grid points in the simulation and hence the computational resources needed for the simula­tion. Current computational resources allow the Reynolds numbers in simulations to reach the order of $10^3 - 10^4$ (the range within which the Reynolds number of our simulations lies, see Appendix A), which are substantially smaller than the typical atmospheric values of $10^8$. Fortunately, relevant turbulence statistics such as variances, covariances and entrainment rates tend
towards Reynolds-number independence once the Reynolds number increases beyond $10^3 - 10^4$ (Dimotakis, 2000; Mellado et al., 2018). When studying the properties of the smallest resolved scales and their substantial effect on large-scale dynamics, DNS eliminates the uncertainty introduced by subgrid-scale models and numerical artefacts, which complicates the interpreta­tion and the comparison of results from different LES models (Mellado et al., 2018). DNS provides results that depend only on the achievable Reynolds number in the simulation and one can perform sensitivity analyses to estimate such dependence. The
disadvantage of DNS is the higher computational cost, since DNS typically utilises higher-order numerical schemes and higher resolutions. Nonetheless, DNS is gaining traction as it approaches a degree of Reynolds number similarity that allows certain extrapolation to atmospheric conditions (Monin and Mahesh, 1998; Jonker et al., 2012, 2013; Waggy et al., 2013; Mellado et al., 2018; van Hooft et al., 2018; Mellado, 2019).

In this work, we opt for DNS because we are studying the variance and covariances of various fields in which entrainment may play an important role. It is known that the smallest resolved scales might become important for these variables in typical
resolutions (Sullivan and Patton, 2011). In addition, with strong emission fluxes, subgrid turbulent parametrisation in LES simulations may induce significant errors near the surface where the tracer is emitted (Vinuesa and Porté-Agel, 2005, 2008). Therefore, here DNS provides an alternative for studying the topic and also a tool for an intercomparison study of the LES results. In Sections 3.1.1 and 3.1.2, we will intercompare the results from our DNS model with the LES results by adopting the
same initial conditions as in Vinuesa and Vilà-Guerau de Arellano (2005) for some of our simulations.

We employed the direct numerical simulation model TLab to perform our computational experiments of turbulent mixing of reactive species in the convective boundary layer. The dynamical part of the simulations performed in this work follows similar settings in Garcia and Mellado (2014) and Van Heerwaarden et al. (2014). In the model, the Navier-Stokes equations





of incompressible fluids in the Boussinesq approximation are solved to calculate the buoyancy and the velocity fields:

$$\nabla \cdot \mathbf{u} = 0,$$

$$\frac{\partial \mathbf{u}}{\partial t} + \nabla \cdot (\mathbf{u} \otimes \mathbf{u}) = -\nabla p + \nu \nabla^2 \mathbf{u} + b\mathbf{k},$$

$$\frac{\partial b}{\partial t} + \nabla \cdot (\mathbf{u}b) = \kappa \nabla^2 b,$$

where $\mathbf{u}(\mathbf{x}, t)$ is the velocity vector with components $(u, v, w)$ in the directions $\mathbf{x} = (x, y, z)$ at time $t$, respectively. $p$ is the modified pressure divided by the constant reference density, and $b(\mathbf{x}, t)$ is the buoyancy, expressed as $b = g(\theta_v - \theta_{v,0})/\theta_v$ . The background buoyancy profile is set as $b_0(z) = N^2 z$, where $N^2$ is the Brunt-Väisälä frequency. The surface buoyancy flux is set to be $F_b$ (see Appendix A). The parameters $\nu$ and $\kappa$ are the kinematic viscosity and the molecular diffusivity respectively. $\mathbf{k}$ is the unit vector in the $z$ direction. The system is statistically homogeneous in the horizontal direction, such that its statistics

depend on $z$ and $t$ only.

A no-penetration, no-slip boundary condition is imposed at the surface and a no-penetration, free-slip boundary condition is imposed at the top boundary. Neumann boundary conditions are imposed for the buoyancy and velocity fields at both the top and the surface to maintain constant fluxes. The velocity and buoyancy fields are relaxed towards zero and $N^2 z$ respectively at the top of the computational domain. The height of the top boundary is adjusted so that the turbulent region is far enough from

the top to avoid significant interaction. Periodic boundary conditions are implemented in lateral directions.

The size of the computational grid is $720 \times 720 \times 512$ for all simulations (the number of vertical layers is 512). Stretching is applied vertically to increase the resolution near the surface in order to resolve the surface layer, and to stretch the grid size in the upper portion of the domain end to further separate the top boundary from the turbulent region. The total simulation domain size is $120L_0 \times 120L_0 \times 34.4L_0$, where $L_0$ is the reference length scale. Together with the reference time and velocity time

scales $T_0$ and $U_0$, $L_0$ is used to non-dimensionalise the Navier-Stokes equations (see Appendix A). With typical atmospheric parameters of $L_0 \sim 100$ m and $U_0 \sim 1$ m s$^{-1}$, the horizontal resolution of the model in use is equivalent to 15 m $\times$ 15 m with a total domain size of 12 km $\times$ 12 km. The resolution is considerably higher than those adopted in the previous LES studies, and is fine enough to resolve convective turbulent motion in the convective boundary layer. In the vertical direction, the grid spacing increases from $\Delta z = 6.72$ m at the surface to $\Delta z = 266.28$ m at the top of the domain at 16 km. The vertical grid

spacing versus height in the first 5 km is shown in Figure 7 (black line).

We use a sixth-order compact scheme to calculate the spatial derivatives, and a forth-order Runge-Kutta scheme to advance the equations in time. The pressure-Poisson equation is solved by applying a Fourier decomposition along the horizontal directions and solving the resultant set of finite-difference equations in the vertical direction to machine accuracy (Mellado and Ansorge, 2012). The sensitivity to grid resolution has been tested up to the triple-velocity correlation in the transport term of

the evolution equation for the turbulence kinetic energy, observing an error of less than 5% at the surface when doubling the spatial resolution. Further details of the numerical algorithm and the grid-sensitivity studies can be found in Mellado (2010), Mellado (2012), Garcia and Mellado (2014) and Mellado et al. (2017).



The simulations are terminated after a total simulation time equivalent to 4.5 hours, during which the boundary layer grows convectively. The simulation time represents the hours from sunrise to midday, when the boundary layer is well developed and statistical equilibrium is attained. The boundary layer height $z_i$ is identified as the height where the buoyancy variance $[b'b']$ is maximum away from the surface layer (Garcia and Mellado, 2014). At the end of our simulations, the boundary layer height is around 2.2 km. As the boundary layer grows in depth, the corresponding convective velocity and time scales evolve in time. In this study, the convective timescale is adopted as the turbulent timescale $t_{turb}$, which is related to the convective velocity $w_c$ by

$$t_{turb} = \frac{z_i}{w_c} = \frac{z_i^{\frac{2}{3}}}{F_b} \tag{1}$$

(Deardorff, 1970). This turbulent timescale will be used to calculate the Damköhler number $Da$ in later sections.

## 2.2 Chemical settings

### 2.2.1 Homogeneous emissions

An archetypical entrainment-emission configuration, as typically used by other LES past studies such as Krol et al. (2000) and Vinuesa and Vilà-Guerau de Arellano (2005), is adopted in our DNS simulation (see left panel of Figure 1). The simulations are conducted with two theoretical tracers A and B. Tracer A is emitted from the surface at a constant flux $F_A$ at a range of values. Tracer B is entrained from the free troposphere, in which the mixing ratio of Tracer B ($\langle B \rangle_0$) is constantly and homogeneously fixed at 2 ppb. Tracer C is formed in a second-order chemical reaction between Tracer A and B:

$$k : A + B \rightarrow C, \tag{R1}$$

where $k$ is the imposed chemical reaction rate. The mixing ratios of the three species, denoted as $A$, $B$ and $C$ respectively, vary with time with rates equal to

$$\frac{dC}{dt} = -\frac{dA}{dt} = -\frac{dB}{dt} = kAB \tag{2}$$

Neumann boundary conditions are imposed at the boundaries of the computational domain on the mixing ratios of the tracers so that the surface flux of Tracer A is constant at $F_A$, and the surface fluxes of B and C are zero. In the free troposphere, the initial mixing ratios of A and C are zero. The initial mixing ratio of Tracer B is decreased linearly down the boundary layer to zero on the surface.

Four emission fluxes of Tracer A ($F_A$), at 0.05 ppb m s$^{-1}$, 0.25 ppb m s$^{-1}$, 0.5 ppb m s$^{-1}$ and 1.41 ppb m s$^{-1}$, are adopted to test the effect of strong emissions in an urban environment. The runs are named VV05 (from its similar initial conditions as Vinuesa and Vilà-Guerau de Arellano (2005)), mflux, sflux and ssflux respectively. As a reference, in the ssflux run, the characteristic mixing ratio scale for Tracer A ($\langle A \rangle_0$) is equal to that of Tracer B ($\langle B \rangle_0$). These characteristic scales are used in the non-dimensionalisation of Equation 2 (see Appendix A). Two chemistry cases, namely slow and fast chemistry, are considered for each of the imposed emission fluxes, with respective chemical reaction rates of $4.75 \times 10^{-4}$ and $4.75 \times 10^{-3}$ ppb$^{-1}$ s$^{-1}$. In the real atmosphere, the rate of chemical reaction between NO and O$_3$ is $4.75 \times 10^{-4}$ ppb$^{-1}$ s$^{-1}$, corresponding to the cases





with slow chemistry. In the meanwhile, the reaction rate between isoprene and OH is 0.1 ppb$^{-1}$ s$^{-1}$ (Karl et al., 2004), giving an example illustrating the cases with fast chemistry. The names and simulation parameters of the eight simulations are listed in Table 1.

### 2.2.2 Heterogeneous emissions

For the simulations with heterogeneous emissions, Tracers A and B are emitted alternately from patches on the surface with widths of 1 km, 2 km and 6 km respectively. The width of these emission patches are referred in this study as the length of heterogeneity $dx$, similarly as in Ouwersloot et al. (2011)[1]. The A-B-C chemical system ($A + B \rightarrow C$) is implemented with slow and fast chemistry, as described in Section 2.2.1. Both Tracers A and B are emitted at the same imposed flux, with the two values of emission fluxes at 0.05 ppb m s$^{-1}$ (VV05) and 0.5 ppb m s$^{-1}$ (sflux) adopted[2]. The right panel of Figure 1 shows a
schematic diagram describing the simulation configuration. The names and simulation parameters of the twelves simulations are listed in Table 1. The simulations with heterogeneous emissions are run for 8 hours, representing the hours from sunrise to the afternoon. These simulations are run for a longer time than those with homogeneous emissions because the statistical equilibrium takes a longer time to be attained with heterogeneous emissions.

In the context of an urban environment, the second-order chemical reaction imposed in our simulations can be considered
analogous to the reaction between NO and peroxyl radical derivatives (RO$_2$) from the oxidation of volatile organic compounds (VOCs), which is the limiting reaction of ozone production in the VOC-limited regime. The VOCs are often emitted from sources segregated from NO at very high fluxes in urban areas. The involved chemical reactions are often relatively fast. For instance, the reaction rate between NO and acetone peroxy radical (CH$_3$COCH$_2$O$_2$), which is a common VOC species found in urban environments (Brasseur and Jacob, 2017; Li, 2019), is 0.503 ppb$^{-1}$ s$^{-1}$ (Manion et al., 2008), giving an example for
the fast-chemistry cases in our simulations with heterogenous emissions.

### 2.3 Quantifying the chemical-turbulence interaction

Two numbers are mainly employed to quantify the effect of turbulent mixing on chemical reaction, namely the Damköhler number $Da$ and the effective chemical reaction rate $k_{eff}$. The first number, the Damköhler number ($Da$), is defined as the ratio between the turbulent timescale ($t_{turb}$) and the chemical timescales ($t_{chem}$) (Damköhler, 1940). For a second-order chemical
reaction $A + B \rightarrow C$, the Damköhler number of Tracer A is given by

$$Da_A = \frac{t_{turb}}{t_{chem,A}} = \frac{z_i/w_c}{1/k\langle B \rangle} = k\langle B \rangle \left( \frac{z_i^2}{F_b} \right)^{\frac{1}{3}}. \tag{3}$$

Similarly the Damköhler number of Tracer B can be written as

$$Da_B = k\langle A \rangle \left( \frac{z_i^2}{F_b} \right)^{\frac{1}{3}}. \tag{4}$$

---

[1]Note the length of heterogeneity referred in this study is equivalent to half of the length denoted in Ouwersloot et al. (2011).
[2]Note that the adopted emission fluxes are doubled from the values in the simulations with homogeneous emissions in order to conserve the total fluxes.



The Damköhler numbers of Tracers A and B at the start ($Da_{A,i}$ and $Da_{B,i}$) and at the end ($Da_{A,f}$ and $Da_{B,f}$) of each of our simulations are listed in Table 2. For a slow chemical reaction of two initially-segregated reactants with the respective $Da \ll 1$, the chemical lifetime of the reactants is long enough for turbulence to mix them in the boundary layer at a rate fast enough that they are homogeneously distributed in a confined volume (e. g., a model grid) in a short time compared to the chemical lifetime. In that case chemical reaction can occur everywhere in that confined volume and the corresponding volumetric-mean production rate of Tracer C can be approximated as the product of $k$ and the volumetric-means of the reactants ($k\langle A\rangle\langle B\rangle$). On the other hand, for fast reactions with the respective $Da > 1$, the reactants are not well mixed by turbulence during the theoretical chemical lifetime of the reactants, such that they can react only in a fraction of the confined volume where turbulent motion brings the two species together. In that case the volumetric-mean of the production rate of Tracer C also depends on the turbulent timescale and would be overestimated by $k\langle A\rangle\langle B\rangle$. Hence, the Damköhler number can indicate whether the effect of turbulent mixing on a chemical reaction is significant in the flow. For instance, the eddy turnover timescale for large-scale eddies in a CBL is typical $10^2 - 10^3$ s, to which the chemical lifetime of $NO_X$ and $O_3$ in the boundary layer is comparable ($10^2 - 10^5$). Therefore, the turbulent motions in the CBL potentially affect any chemical reaction occurring at a rate higher than that between NO and $O_3$.

The second number, the effective chemical reaction rate ($k_{eff}$), can quantitatively measure the actual chemical reaction rate averaged in a confined volume under the consideration of chemical segregation due to inefficient turbulent mixing (Vinuesa and Vilà-Guerau de Arellano, 2005), and can be written as

$$k_{eff} = \left(1 + \frac{\langle A'B'\rangle}{\langle A\rangle\langle B\rangle}\right)k, \tag{5}$$

where the angle-bracketed and dashed terms are the means and the deviations from the means respectively, of the Reynolds-decomposed concentrations.

The error induced by neglecting such chemical segregation in a complete-mixing model, in which both Tracer A and B are assumed to be evenly distributed throughout the boundary layer, can be written as

$$E = 1 - \frac{k_{eff}}{k} = -I_S, \tag{6}$$

where the segregaton coefficient

$$I_S = \frac{\langle A'B'\rangle}{\langle A\rangle\langle B\rangle} \tag{7}$$

represents the state of mixing of Tracers A and B in the boundary layer (e. g. Danckwerts (1952); Vinuesa and Vilà-Guerau de Arellano (2003)). If Tracers A and B are completely mixed, then $I_S = 0$. If Tracers A and B are completely segregated, then $I_S = -1$. If Tracers A and B are emitted simultaneously and at the same location so that their concentrations are correlated, then $I_S > 1$. With the DNS model fully resolving the chemical segregation in the boundary layer, the error from a complete-mixing or coarse-grid model can be calculated by

$$E = k_{eff,\ cm} - k_{eff,\ DNS}, \tag{8}$$





where $k_{eff,\,cm}$ is the calculated boundary layer-averaged effective chemical reaction rate in a complete-mixing or coarse-grid model, and $k_{eff,\,DNS}$ is the calculated value in the corresponding DNS model.

When one considers a confined volume within a horizontal layer at a specific height $z$, the corresponding effective chemical reaction rate also varies with height. The height-dependent effective chemical reaction rate can be written as a function of $z$:

$$[k_{eff}](z) = (1 + \frac{[A'B']}{[A][B]})k = (1 + [I_S](z))k. \tag{9}$$

The square-bracketed terms denote the horizontal means of the quantities. The corresponding height-dependent error induced by neglecting chemical segregation by a model that assumes Tracers A and B are well-mixed horizontally within a horizontal layer at $z$ is then

$$[E](z) = 1 - \frac{[k_{eff}](z)}{k} = -[I_S](z), \tag{10}$$

where the height-dependent segregation coefficient is

$$[I_S](z) = \frac{[A'B']}{[A][B]}. \tag{11}$$

## 3 Results of the DNS simulations

### 3.1 Homogeneous emissions

#### 3.1.1 Reference cases with rural emission flux

For both VV05 runs, all the tracers are relatively well-mixed in the mixed layer. Tracer A is largely consumed in the mixed layer, while Tracer B is in excess so that its concentration remains essentially constant. The reaction between Tracers A and B can then be considered as a pseudo first-order reaction, and the corresponding production term $kAB \approx k'A$ (plotted in the top panel of Figure 2) is dependent only on the concentration of Tracer A, where $k' = kB$ is the pseudo first-order rate coefficient (Hobbs, 2000; Jacobson and Jacobson, 2005). In this situation, Tracer A is said to be the limiting reactant, and the chemical reaction between Tracers A and B is Tracer A-limiting (Zumdahl, 1992). As the production of Tracer C is also Tracer A-limiting, the vertical fluxes of Tracer C (the magenta lines in Figure 3) in these two cases are both positive, similar to those of Tracer A (not shown), indicating that Tracers A and C are both correlated with the updrifting air. As the chemistry shifts from slow to fast, the height with the maximum flux of Tracer C transits from near the top of the boundary layer to near the surface. This is due to the increasing lifetime of Tracer B with an increasing chemical reaction rate. As more Tracer A is consumed, more Tracer B can flow down the boundary layer without being consumed, and hence more Tracer C is produced at a lower altitude in the boundary layer where Tracer A is available. As indicated by a large Damköhler number of Tracer A ($Da_{A,f} = 5.71 > 1$), the mixing between Tracer A and B hence becomes less efficient, resulting in a larger segregation between the two tracers.





### 3.1.2 Comparison with previous LES studies

We use the results of the two VV05 runs to compare with those presented in the LES study of Vinuesa and Vilà-Guerau de Arellano (2005). The horizontal and vertical resolutions of their LES model are 50 m and 25 m respectively, roughly 4 times coarser than the resolutions of our DNS model. From our DNS model, the resultant boundary layer-averaged effective chemical

reaction rates $k_{eff}$ are 96.5% and 86.9% of the imposed rate $k$ for our slow-VV05 and fast-VV05 runs respectively (see last column of Table 2). These rates are also similar to the values of 96% and 85% reported in Vinuesa and Vilà-Guerau de Arellano (2005). Therefore, our DNS model gives similar boundary layer-averaged values of $k_{eff}$ as the LES model adopted in Vinuesa and Vilà-Guerau de Arellano (2005). When comparing our vertical profiles of the horizontally-averaged effective chemical reaction rate (Figure 4) with the vertical profiles of the segregation coefficient presented in Vinuesa and Vilà-Guerau de Arellano

(2005), the profiles of the two studies in general share similar shapes, with the largest magnitude of segregation all occurring just below the top of the boundary layer. For instance, the maximum values of $[k_{eff}]$ are 83% and 62% of $k$ in our slow-VV05 and fast-VV05 runs respectively[3]. However, our profiles show more prominent minima near the top of the surface layer, which shows the effect of the higher vertical resolution in the DNS model.

### 3.1.3 The effect of strong emission fluxes

When the emission fluxes of Tracer A increase to urban values beyond 0.25 ppb m s$^{-1}$ in the mflux, sflux and ssflux runs, Tracer A accumulates and is in excess in most of the boundary layer. On the other hand, Tracer B is almost completely depleted in the lower part of the boundary layer. As the emission flux of Tracer A increases and/or the chemistry becomes faster, the chemical timescale of Tracer B shortens, so that the depth where Tracer B can be replenished from the free troposphere becomes narrower. Tracer C is mostly produced near the top of the boundary layer, as indicated in the colour map of the production

term in the middle panel of Figure 2. The chemical system now transits to the so-called "diffusion-limited" reaction (Sykes et al., 1994), where the reaction between Tracers A and Tracer B depends on the availability of Tracer B, and hence is Tracer B-limiting.

The shift of the reaction from Tracer A-limiting to Tracer B-limiting can also been seen from the profiles of the vertical fluxes of Tracer C on Figure 3 (sflux in red and ssflux in cyan). The fluxes of Tracer C now become negative, indicating that

Tracer C is downdraft-correlated, as is Tracer B. As the emission flux of Tracer A increases, the maximum of the production term shifts from the top of the surface layer to near the top of the boundary layer closer to the source of Tracer B. These cases are also characterised by very large values of $Da_{B,f}$, instead of a large $Da_{A,f}$ as in the fast-VV05 run (see Table 2). Since the chemical transformation of the reactant that is relatively less abundant than the other reactant is influenced more by turbulent mixing (Vilà-Guerau de Arellano et al., 2004), $Da_{B,f}$ is now a better indicator for the role of turbulent mixing on

chemical reactions than $Da_{A,f}$. Summarising the simulations with homogeneous emissions, one can observe from Figure 8

---

[3]Unfortunately we cannot compare the vertical profiles of our horizontally-averaged effective chemical reaction rate with the vertical profiles of the segregation coefficient presented in Vinuesa and Vilà-Guerau de Arellano (2005) quantitatively, as the latter profiles are from the LES studies in Vinuesa and Vilà-Guerau de Arellano (2003), which adopted a different set of initial conditions as Vinuesa and Vilà-Guerau de Arellano (2005) and our study.





(black circles) that the deviation of $k_{eff}$ from the imposed rate $k$, or the error from the complete-mixing model, increases with the increased Damköhler number of the limiting reactant ($Da_{lim}$), where $Da_{lim} = Da_{A,f}$ when the reaction is Tracer A-limiting and $Da_{lim} = Da_{B,f}$ when the reaction is Tracer B-limiting.

For the runs with urban emission fluxes, the boundary layer-averaged effective chemical reaction rate $k_{eff}$ is very low even with slow chemistry. With slow chemistry, $k_{eff}$ is only 23.5%, 12.3% and 8.2% of the imposed rate $k$ in the mflux, sflux and ssflux runs respectively. With fast chemistry, $k_{eff}$ further drops to 4.6%, 3.0% and 2.3% of $k$ respectively. These low values of $k_{eff}$ imply that if the surface emission of a pollutant is so strong that the reaction is limited by the availability of the entrained reactant, the resultant overestimation of the actual chemical reaction from an assumption of complete mixing can be substantial (in our cases up to 98%) for any reaction with reaction rate equivalent to or faster than that between NO and $O_3$. The vertical profiles of the horizontally-averaged effective chemical reaction rate in Figure 4 show that the maximum segregation between Tracer A and B still occurs near the top of the boundary layer. With fast chemistry, the effective chemical reaction rate at that altitude can even drop to around 10% of the imposed rate.

## 3.2 Heterogeneous emissions

With the emissions of Tracers A and B heterogeneously distributed on the surface, the segregation is maximum at the surface with its magnitude decreasing with increasing altitude, as seen from the vertical profiles of the horizontally-averaged segregation coefficient ($[I_S](z)$) in Figure 5. It is because under this setting, mixing is most difficult near the surface where the tracers are emitted separately. This description of the profiles agrees with the results of the simulations with heterogeneous emissions presented in Krol et al. (2000) and Vinuesa and Vilà-Guerau de Arellano (2005), despite that they implemented a different emission configuration with a Gaussian function imposed on the surface emission flux. To examine how the segregation between Tracers A and B changes with the emission flux and the imposed chemical reaction rate, the results of the set of simulations with the length of heterogeneity ($dx$) of 2 km (all plotted in red) are first compared. In the slow-2km-VV05 run (red dashed line), the magnitude of $[I_S]$ decreases with increasing altitude in the boundary layer, so that at a height above $0.8z_i$, $[I_S]$ becomes larger than zero, meaning that Tracers A and B are not only well-mixed but their flows are also correlated. This results in $[k_{eff}] > k$, where the assumption of complete mixing in the model grid will underestimate the reaction rate. This phenomenon is also reported in Ouwersloot et al. (2011), showing $[I_S] > 0$ at higher altitudes in their LES simulations with heterogeneous isoprene emission fluxes. On the other hand, as the surface emission flux increases to 0.5 ppb m s$^{-1}$ in the slow-2km-sflux run (red solid line), $[I_S] > 0$ only at the height above $0.98z_i$.

When shifting from slow to fast chemistry (red dotted line), the magnitude of the segregation further increases, resulting in a boundary layer-averaged value of -0.95. The segregation at the top of the boundary layer remains high, with $[I_S] = -0.8$. Sometimes the segregation caused by an increase of imposed reaction rate $k$ can reduce $k_{eff}$ so much, that merely increasing $k$ can no longer increase the production of Tracer C. This happens between the two 6km-sflux runs (check Table 2 for their resultant $k_{eff}$). When shifting from slow chemistry to fast chemistry, $k_{eff}$ drops 9.18 times, such that it in turn cancels the effect of the 10-time increase in $k$. This results in similar amounts of Tracer C produced in the two 6km-sflux runs, with the





mixing ratio of Tracer C equal to 1.58 ppm and 1.60 ppm at the end of the simulations for the slow- and fast-chemistry cases respectively.

To discuss the effect of the length of heterogeneity ($dx$) on segregation, the three solid lines in the left panel of Figure 5, which indicate the slow-sflux runs at different $dx$ ($dx = 1$ km plotted in blue, $dx = 2$ km in red and $dx = 6$ km in green), are

compared. In general, the segregation increases with increasing $dx$. This agrees with the conclusion from Ouwersloot et al. (2011). For the cases with $dx = 1$ km and $dx = 2$ km, one can still observe a decrease of the magnitude of the segregation with increasing altitude, with the corresponding boundary layer-averaged $I_S$ equal to -0.49 and -0.69 respectively. However, for $dx = 6$ km, the segregation remains almost constant throughout the mixed layer, and starts to decrease only below the top of the boundary layer. This also results in a very large segregation throughout the boundary layer, with the boundary layer-

averaged $I_S$ equal to -0.97. This shows similar phenomenon as described in Ouwersloot et al. (2011), that as the length of heterogeneity exceeds a few boundary layer height[4], i. e. $\sim 3$ km at the end of the simulations, the boundary layer between the two patches that respectively emit Tracers A and B barely mix, and the system behaves like two individual boundary layers.

Similar to the cases with homogeneous emissions, the increased segregation with increasing surface emission flux and imposed chemical reaction rate can be indicated by the relatively large values of the final Damköhler numbers. Since the boundary

layer-averaged Damköhler numbers of Tracers A and B are statistically the same in the simulations with heterogeneous emissions, here we take the mean of the two numbers to get the averaged final Damköhler number $Da_f$ ($= 1/2(Da_{A,f} + Da_{B,f})$). We then fit $Da_f$ with the normalised boundary layer-averaged $k_{eff}/k$ for the simulations with each $dx$ (check Table 2 for the values). The results are plotted in Figure 6. In general, $\log(k_{eff}/k)$ follows a square law with $\log(Da_f)$, i. e. $\log(k_{eff}/k) = -a_{fit}(\log Da_f + 1)^2$, where $a_{fit}$ is the fitting coefficient varying with $dx$. Or in another way, $k_{eff}/k$ follows a Gaussian func-

tion of $Da_f$, or $k_{eff}/k = \exp(-a_{fit}(\log Da_f + 1)^2)$. The x-intercept is chosen to be -1 because $k_{eff} \sim k$ when $Da_f \lesssim 0.1$ (or $\log Da_f \sim -1$). As $dx$ increases, $a_{fit}$ also increases (see the values in Figure 6). Their relation is however unlikely to be linear.

## 4 Estimating errors from regional models

In the previous sections, the boundary layer-averaged effective chemical reaction rate is compared with the imposed rate in a complete-mixing model that assumes the tracers to be completely well mixed in the whole boundary layer. However,

the horizontal resolutions in regional chemical-transport models are comparatively high in order of a few kilometres with multiple vertical levels within the boundary layer. Such models are often employed when modelling urban areas. To evaluate the importance of the subgrid chemistry-turbulence interaction in these regional chemical-transport models, we degrade our DNS model to coarse-grid models with two horizontal resolutions (1-km and 3-km, with 12 and 4 nodes in the horizontal direction respectively) and two vertical resolutions (32-lev and 64-lev, with 32 and 64 nodes in the vertical direction in the

whole domain, and with 10 and 20 levels within the boundary layer respectively when the boundary layer is fully grown), resulting in four coarse-grid models, named 1 km-64 lev, 1 km-32 lev, 3 km-64 lev and 3 km-32 lev. Figure 7 shows the

---

[4]Converting the conclusions of Ouwersloot et al. (2011) to our denotation, the separation of the boundary layer above the different emission patches occurs when $dx > 8z_i$.





respective vertical resolutions of the DNS model and the 64-lev and 32-lev coarse-grid models versus height. These resolutions are selected according to common grid sizes in regional chemical-transport models (e. g. Bouarar et al. (2017); Marécal et al. (2015)).

The tracer concentration fields are taken from the corresponding DNS simulations, and interpolated onto the grids of the respective coarse-grid models. The volumetric averages of tracer concentrations in each model grid are then calculated. The statistics of these resolution-degraded concentration fields are then calculated as in Section 3. The chemical production terms from a coarse-grid model and the corresponding DNS simulation are plotted in the top and bottom panel of Figure 2 respectively to illustrate the effect of resolution degrading. In contrast to the DNS model, the coarse-grid model can no longer resolve the chemical segregation within each of its model grids. The boundary layer-averaged normalised effective chemical reaction rates $k_{eff}/k$ of the four coarse-grid models averaged over the last 1.5 hours of the corresponding simulations are subtracted from the values of $k_{eff}/k$ from our DNS model (last column of Table 2) to obtain the boundary layer-averaged errors from the coarse-grid models induced by neglecting the subgrid chemical segregation ($E$) (see Equation 8). These errors are listed in Table 3. The second column is the error from a complete-mixing model ($E_{cm}$), calculated from Equation 6. Note that the errors are listed in percentages. Positive values indicate overestimation of the model from the actual chemical reaction rate, and negative values indicate underestimation. The dominance of positive values shows that the coarse-grid models, similar to the complete-mixing model, usually overestimate the actual chemical reaction rate.

## 4.1 Homogeneous emissions

We take a closer look at the results of the simulations with homogeneous emissions as described in Section 3.1. Figure 8 shows the model errors from the complete-mixing model (black circles) and the four coarse-grid models (+ and × marks) against the corresponding final Damköhler number of the limiting reactant ($Da_{lim}$, see the values in Table 2). One can see that in all runs the errors from all coarse-grid models are reduced in comparison to the complete-mixing model. The model errors from the coarse-grid models are largely reduced when $Da_{lim} > 10$, which corresponds to the simulations with urban surface emission fluxes (mflux, sflux and ssflux). In those runs, the model errors drop from 77 - 98 % in the complete-mixing model, to 36 - 69% in the 32-lev models, and further to 15 - 51% in the 64-lev models. However, this also means that even with the highest resolution the model errors are still noticeably significant. Also, for 5 out of 7 runs (the slow-VV05 run is excluded here due to the insignificant errors), a larger reduction of the model error is observed with an increased vertical resolution (from blue + to blue ×) than with an increased horizontal resolution (from blue + to red +). The reason behind is that Tracers A and B are segregated along the vertical direction.

Figure 9 shows the vertical profiles of the horizontally-averaged error ($[E](z)$, see Equation 10) of the four coarse-grid models for the slow-mflux and slow-sflux runs. All coarse-grid models fail to show the local minimum at the top of the surface layer, as the surface layer is too thin for these models to resolve. They also show a less prominent minimum at the top of the boundary layer. The distinct underestimation of $[E]$ in the surface layer and entrainment zone indicates that high vertical resolution is especially important to resolve the chemical segregation around these two zones.





## 4.2 Heterogeneous emissions

The errors from the four coarse-grid models ($E$) in the simulations with heterogeneous emissions are then plotted against the errors from the corresponding complete-mixing model ($E_{cm}$) in Figure 10. These data points are categorised with different length of heterogeneity ($dx$) implemented in the corresponding simulations. The dashed line indicates the value where $E =$
$E_{cm}$ . The points below the dashed line show the cases for which the corresponding coarse-grid model improves from the complete-mixing model, while those above the dashed line show a deterioration. Contrary to the simulations with homogeneous emissions, the coarse-grid models do not always perform better than the complete-mixing model, especially when the horizontal model grid size is larger than the length of heterogeneity. It is because under this condition, the model grid fails to resolve the heterogeneity of the emission sources. This also applies to the simulations with $dx = 1$ km in the 1 km-resolution coarse-
grid models, as the model grids are slightly offset from the emission patches. In these runs, the coarse-grid models mix the originally-segregated tracers in the lower layers at a higher concentration than the complete-mixing model, in which the latter artificially mixes the tracers all over the whole boundary layer and dilutes the tracer concentrations. That is why the coarse-grid models give larger overestimations. The model with the higher vertical resolution overestimates the reaction rate even more, as its first layers are thinner, and hence contain even higher concentrations of the artificially-mixed tracers in the lower-level
grids.

     For the simulations with $dx = 6$ km, the coarse-grid models always perform better than the complete-mixing model. In these simulations, the horizontal grids in the coarse-grid models can always resolve the emission heterogeneity. Note that the reduction of the errors from the coarse-grid models is larger for the increased horizontal than vertical resolution, as Tracers A and B are now segregated in the horizontal direction.

## 20   5   Discussions

### 5.1   Implications for regional modelling of urban environments

Although the DNS simulations in this work are only conducted in idealised conditions, they can provide insights and estimations of the errors induced by neglecting the subgrid chemical segregation due to inefficient turbulent mixing in larger-scale models. The simulations with homogeneous emissions (Section 3.1) show significant errors in the low-resolution model under
strong emission flux conditions, which are indicated by a large ($> 1$) Damköhler number of the limiting reactant ($Da_{lim}$). This suggests that $Da_{lim} > 1$ can be taken as a condition in the model under which the chemical calculation should be corrected by taking the subgrid chemical segregation into account. One way to implement such correction is to substitute the imposed chemical reaction rate $k$ by the effective rate $k_{eff}$ which, as shown in Figure 8, can be expressed as a function of $Da_{lim}$. The results from the simulations with heterogeneous emissions (Figure 6) suggest that $k_{eff}$ additionally depends on the length of
heterogeneity ($dx$). When considering increasing model resolution to reduce the errors induced by neglecting subgrid chemical segregation, we learn from Section 4 that whether it is more effective to increase the horizontal or vertical resolution depends on the direction of the initial segregation of the reactant sources. High resolution is in particular necessary around the surface





layer and entrainment zone. When the reactant sources are horizontally segregated, it is essential that the horizontal resolution of the model is fine enough to resolve the emission heterogeneity. Otherwise, increasing the vertical resolution can induce even larger model errors.

There are additional points to note in our estimations. When arriving our conclusion of the dependency of $k_{eff}$ on $Da_{lim}$, we increase $Da_{lim}$ by shortening the corresponding chemical timescale with increased imposed $k$ and emission fluxes. One should not neglect the dependency of $Da_{lim}$ on the buoyancy fluxes, which determine the turbulent timescale. However, we expect to see the similar $k_{eff}$ dependency on $Da_{lim}$ when one repeats the simulations with increasing buoyancy fluxes instead.

By increasing the emission fluxes to urban values in Section 3.1.3, it is shown that the effective chemical reaction rate can be much smaller than the values reported in previous studies under rural conditions (e. g. Vinuesa and Vilà-Guerau de Arellano (2003, 2005)). However, it should be also noted that the chemical segregation is particularly large in these simulations because one of the reactants (in this case Tracer B) is depleting. Despite the large deviation of $k_{eff}$ from $k$, the production term $kAB$ is anyway small due to the low concentration of Tracer B. Therefore the actual effect on the calculated concentration of Tracer B may not be as large as one may expect from the overestimation of $k_{eff}$. A similar reasoning also applies to Tracer A. When the emission flux of Tracer A is large, the main process affecting the concentration of Tracer A is emission instead of chemistry. However the effect of chemical segregation on the concentrations of the product Tracer C is still worth noticing.

Unlike other studies (e. g. Ouwersloot et al. (2011); Dlugi et al. (2019); Kim et al. (2016); Li et al. (2016, 2017)) in which multiple-reaction chemical systems are employed, our work mostly focuses on an idealised second-order chemical reaction of two non-specific reactive species. This approach allows us to interpret our work for any second-order chemical reactions. For a chemical species involved in a multiple-reaction chemical system, like $O_3$, one can still calculate the net impact of chemical segregation by considering the errors of all reactions in which the species is involved. However, it is also important to notice that the net impact of chemical segregation on such a species would in general be reduced with the increasing complexity of the chemical system, because cycling reactions tend to lengthen the chemical timescale and reduce the corresponding Damköhler number, and hence reduce the chemical segregation. For example, with emission fluxes of $NO_X$ comparable to our sflux and ssflux cases, the segregation coefficient between isoprene and OH is only -0.05 and -0.17 in the high- and very high-$NO_X$ cases of Kim et al. (2016) respectively. Therefore we expect the magnitude of chemical segregation to be smaller than the values reported in this study when a multiple-reaction chemical system is included.

The work in Section 4 gives an estimation of the error in a regional chemical-transport model when neglecting subgrid chemical segregation, where the model possesses relatively high horizontal and vertical resolution. The method we employed is to average the concentration fields within the volume of each model grid. Current regional chemical-transport models often include additional measures to prevent accumulation of reactants close to the emission sources, such as distributing the emitted species in first few vertical layers (e. g. Freitas et al. (2011)) and employing boundary layer schemes to parametrise eddy transport in the lower troposphere (e. g. Hong et al. (2006)). These measures may diminish the excessive overestimation in the coarse-grid models reported in Section 4.2 in cases of which the model fails to resolve emission heterogeneity. However, these measures may introduce additional artificial mixing in other cases. Also, there is an inherited difference in modelling framework that makes the comparison between regional chemical-transport models and turbulence-resolving models like DNS





and LES difficult. The deficiency of regional models to only parametrise the vertical mixing in the boundary layer causes its inaccuracy in representing the boundary layer dynamics. For instance, by comparing the model results from WRF-Chem and from the NCAR LES model, Li et al. (2019) reported a weaker vertical mixing in the WRF-Chem simulations than in the LES simulations, causing the upward transports of surface-emitted chemicals and hence the chemical production of OH and

$O_3$ to be undermined aloft in the mixed layer. Our averaging method employed in the coarse-grid models may not be able to account for these undermined upward transports in regional chemical-transport models. Nevertheless, our estimation shows that the model resolution clearly plays an important role on resolving chemical segregation and providing accurate chemical calculations in the models.

One important aim of the study of chemistry-turbulence interaction is to provide a correction to the error in large-scale model

induced by neglecting subgrid chemical segregation. While our work suggests a correction of $k_{eff}$ with dependencies on $Da_{lim}$ and $dx$, other work suggest that such correction should also include other variables such as updraft/downdraft fluxes (Petersen and Holtslag, 1999), variance of the reacting species, entrainment/emission fluxes (Petersen and Holtslag, 1999; Vinuesa and Vilà-Guerau de Arellano, 2003), turbulent fluxes (Dlugi et al., 2014), variance of the emission (Galmarini et al., 1997), magnitude and direction of mean horizontal wind (Ouwersloot et al., 2011) and distance from the sources (Karamchandani et al.,

2000). There have been attempts to implement a parametrisation of subgrid chemistry-turbulence interaction to large-scale models. For example, Lenschow et al. (2016) developed a one-dimension second-order closure model to account for the vertical turbulent mixing of chemical species ready to be incorporated into large-scale models. Given that the effect of subgrid chemical segregation is non-negligible under urban conditions, modellers should consider applying similar parametrisations in areas with intense emission and large source heterogeneity.

**5.2   Other factors to be considered when studying chemical segregation in urban environments**

Due to computational limitations of the DNS simulations, some other conditions that may be important for turbulent mixing in the urban boundary layer were neglected in our work. First of all, the growth of the boundary layer is driven by a constant buoyancy flux, so that the boundary layer height gradually increases. Hence, the simulation can only approximate the time when the boundary layer grows from sunrise to mid-afternoon. This also raises the issue of the time required for statistical

equilibrium to attain. In some cases, this time may be longer than the duration of daylight, after which the atmosphere is no longer convective. This poses a greater problem to cases with strong emission fluxes, as the time required to attain statistical equilibrium is even longer. It may not be practical to run longer than the duration of daylight even though statistical equilibrium is not reached. Second, our simulations only address a convective boundary layer under clear-sky conditions. We do not take other scenarios with different weather conditions (such as cloud-top boundary layer, e. g. Li et al. (2016, 2017)) or when the

atmosphere is stably stratified (typical for nighttime, e. g. Galmarini et al. (1997); Geyer and Stutz (2004)) into account. Third, it may be useful to consider other boundary conditions of the entrained tracers, such as varied entrainment fluxes (e. g. Krol et al. (2000); Albrecht et al. (2016)) and varied concentrations in the free troposphere due to long-range transport (e. g. Zyryanov et al. (2012)). Fourth, our simulations also do not include any mean horizontal winds (e. g. Ouwersloot et al. (2011)), or forcings from the surface other than the constant buoyancy flux, such as varied surface heat fluxes (e. g. Ouwersloot et al.





(2011); Van Heerwaarden et al. (2014)) and surface roughness. An important source of surface forcings in an urban boundary layer is undoubtedly from the urban structures (buildings and streets in the urban canopy). The structure of turbulent flow can be significantly altered in the street canyons due to the exchange of heat and momentum with the urban structures (e. g. Oke (1997)). Therefore, our DNS simulations are more suitable in addressing the vertical mixing caused by the turbulent motions in

the mixed layer relatively far away from the surface features that may induce other additional forcings. But in the mixed layer, the urban canopy still potentially affects the chemistry and dynamics in the boundary layer by means of surface roughness and emission heterogeneity. While in this work we have addressed the effect of emission heterogeneity, the effect of surface roughness and other configurations of emission patterns (e. g. Auger and Legras (2007)) can be further studied in the future.

## 6  Conclusions

We explore in this work the effect of chemical segregation due to inefficient turbulent mixing on chemical reactions in an urban boundary layer, and estimate the resultant error in the regional chemical-transport models by conducting direct numerical simulations (DNS). As past studies mainly examined scenarios in forestal areas, we focus on urban conditions, specifically with strong emission fluxes and heterogeneous emissions, as both factors can potentially increase chemical segregation.

With homogeneous emissions, our simulations give similar results as past studies using large-eddy simulations when the
emission flux of the surface-emitted Tracer A is of rural value, in spite of the increase in resolution of our DNS model. On the other hand, increasing the emission flux of Tracer A to urban values depletes the entrained Tracer B, so that its availability limits the reaction. In this situation, the segregation between Tracer A and B becomes very large, which results in significant overestimation of the effective chemical reaction rate $k_{eff}$ in a complete-mixing model. Such cases are always characterised by a very large Damköhler number of Tracer B, the limiting reactant, suggesting the Damköhler number of the limiting reactant
$Da_{lim}$ can be used as an indicator of the effect of turbulent mixing on chemical reaction. When $Da_{lim} > 1$, chemical calculations in low-resolution models should be corrected by taking chemical segregation into account, and $k_{eff}$ can be used for such a correction, which can be expressed as a function of $Da_{lim}$. From our simulations with heterogeneous emissions, in which Tracers A and B are emitted from the surface in alternate patches of different widths, $k_{eff}$ normalised by the imposed rate $k$ is further found to be in a Gaussian function of the final Damköhler number of the reactants, with the slope of the fitted Gaussian
law increasing with the length of heterogeneity. The large overestimation of $k_{eff}$ reported from some cases of our simulations shows that the segregation between heterogeneously distributed surface-emitting pollutants is potentially important and should not be neglected especially when the corresponding emission flux and/or the length of heterogeneity are large.

To evaluate the errors induced by neglecting the chemical segregation in regional chemical-transport models with higher resolution than a complete-mixing model, we degraded our DNS model to coarse-grid models with two horizontal and vertical
resolutions commensurable to regional models. With homogeneous emissions, all the coarse-grid models give smaller errors than the complete-mixing model. Yet the errors from the coarse-grid models remain high for simulations with urban emission fluxes. The improvement is more significant for the increased vertical resolution instead of horizontal resolution, as the initial segregation between Tracers A and B is in the vertical direction. All coarse-grid models give the largest overestimations of the





height-dependent effective chemical reaction rate near the top of the surface layer and the entrainment zone, indicating that high resolution is most important in these areas. With heterogeneous emissions, the coarse-grid models perform worse than the complete-mixing model when the coarse horizontal model grid cannot resolve the emission heterogeneity. With higher vertical resolution, the respective coarse-grid model gives an even larger error. This illustrates that increasing the model resolution

may not improve the model performance when the enhanced model resolution still fails to resolve the emission heterogeneity. For the coarse-grid models which can resolve the emission heterogeneity, the model improvement of the coarse-grid model is more significant for increased horizontal than vertical resolution, as in these cases the initial segregation is in the horizontal direction. This suggests that whether the model improvement is more sensitive to the increase in the horizontal or vertical resolution depends on the direction of initial segregation between the reactants.

*Code and data availability.*  Source files of the DNS model TLab and further documentation can be found at https://github.com/turbulencia/tlab (accessed 07 April 2020). Primary data and scripts used in the analysis and other supplementary information that may be useful in reproducing the author's work are archived by the Max Planck Institute for Meteorology and can be obtained at http://hdl.handle.net/21.11116/0000-0006-11C3-A.

**Appendix A:  Non-dimensionalisaion of the model**

Our results from the DNS simulations are based on the data in the fully developed turbulent regime in the simulation that is established after the initial transient phrase. In that regime, the initial conditions have been sufficiently forgotten, and the parameters $\{\nu, \kappa, F_b, N\}$ define the system completely. $N$ and $F_b$ are chosen to non-dimensionalise the equations (Fedorovich et al., 2004; Garcia and Mellado, 2014), such that the system yields a reference time scale $T_0$ as $N^{-1}$, a reference length scale $L_0$ as $(F_b/N^3)^{\frac{1}{2}}$ and a reference velocity scale $U_0$ as $(L_0 F_b)^{\frac{1}{3}}$. With the Prandtl number ($Pr = \nu/\kappa$) set to 1, the system only

depends on the reference buoyancy Reynolds number

$$Re_0 = \frac{F_b}{\nu N^2},$$

which is set to 15000 in all simulations. $Re_0$ refers to the Reynolds number in the entrainment zone (Garcia and Mellado, 2014). In this configuration, the Reynolds number is ~1000 in the mixed layer. Conducting DNS with $Re$ in typical atmospheric condition ($\sim 10^7 - 10^8$) is computationally impossible. However, with Reynolds number similarity, one can perform DNS

with moderate values, which still allows certain extrapolation of the DNS results to corresponding atmospheric conditions (Dimotakis, 2000; Monin and Mahesh, 1998). Reynolds number similarity is an experimental observation of which properties of turbulent flows become practically independent of $Re$ (Tennekes et al., 1972; Mellado et al., 2018). With the values in our simulations, the Reynolds number in the whole boundary layer lies within the range where the Reynolds number similarity applies. One can refer to Section 2.1 for the more detailed explanation of the Reynolds number in DNS models.

The rate equations of the species are non-dimensionalised by introducing the characteristic scales for the mixing ratios. The characteristic scale for Tracer A is controlled by its emission flux $F_A$, and is therefore defined as $\langle A \rangle_0 = F_A U_0^{-1}$. For Tracer



B, the characteristic scale is set as its background concentration in the free troposphere ($\langle B \rangle_0$). For Tracer C, the characteristic scale is defined as $\langle C \rangle_0 = L_0 U_0^{-1} k \langle A \rangle_0 \langle B \rangle_0$ to non-dimensionalise the concentration of Tracer C with its dimensionless production term. The mixing ratios of the tracers are normalised with these scales so that the normalised mixing ratios are defined as $c_A = A/\langle A \rangle_0$, $c_B = B/\langle B \rangle_0$ and $c_C = C/\langle C \rangle_0$. With these definitions, the rates of the three tracers can be expressed as the following equations:

$$L_0 U_0^{-1} \frac{dc_A}{dt} = -K_A c_A c_B$$
$$L_0 U_0^{-1} \frac{dc_B}{dt} = -K_B c_A c_B$$
$$L_0 U_0^{-1} \frac{dc_C}{dt} = -c_A c_B,$$

where the dimensionless rate constants are $K_A = L_0 U_0^{-1} k \langle B \rangle_0$ and $K_B = L_0 U_0^{-1} k \langle A \rangle_0$.

*Author contributions.* CWYL wrote this article, designed the research, conducted the simulations and performed the result analysis of this work; GPB and HS provided guidance and supervision to CWYL and contributed ideas to the work; JPM provided the DNS model TLab, added the chemical tracers into the original DNS model, provided technical support on modelling issues and technical information of the DNS model in this article; In addition, GPB, HS and JPM contributed to the editing of this article.

*Competing interests.* The authors declare that they have no conflict of interests.

*Acknowledgements.* This work has been financially supported by Max Planck Institute for Meteorology as part of the doctoral thesis of CWYL (Li, 2019). The computation of the simulations presented in this work was completed on the Mistral supercomputer of DKRZ. DKRZ also contributed to the storage of the data presented in this work. The article processing charges for this open-access publication were covered by the Max Planck Society. The authors also thank Mary Barth in the National Center for Atmospheric Research for her comments to this article.



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





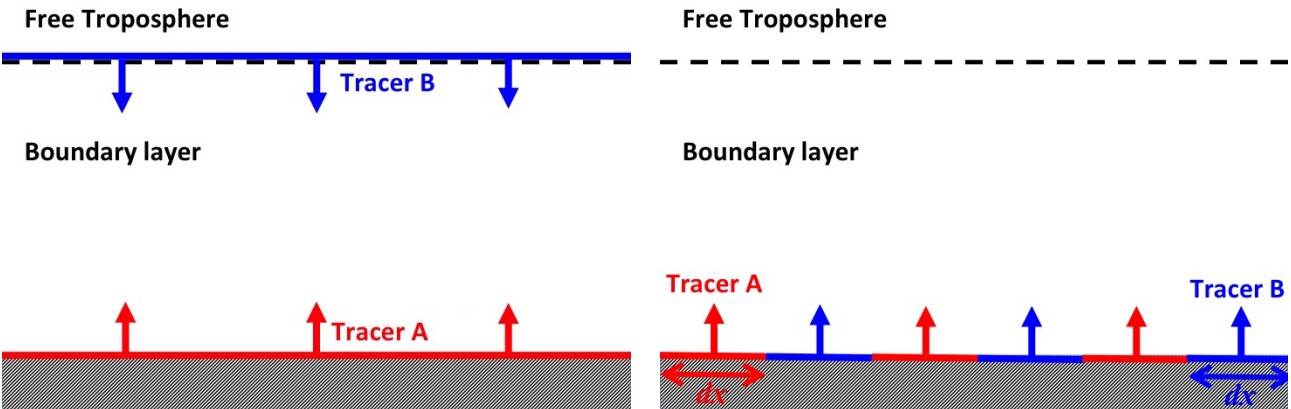

**Figure 1.** Schematic diagrams of the configurations of the DNS simulations with homogeneous emissions (*left*) and heterogeneous emissions (*right*).

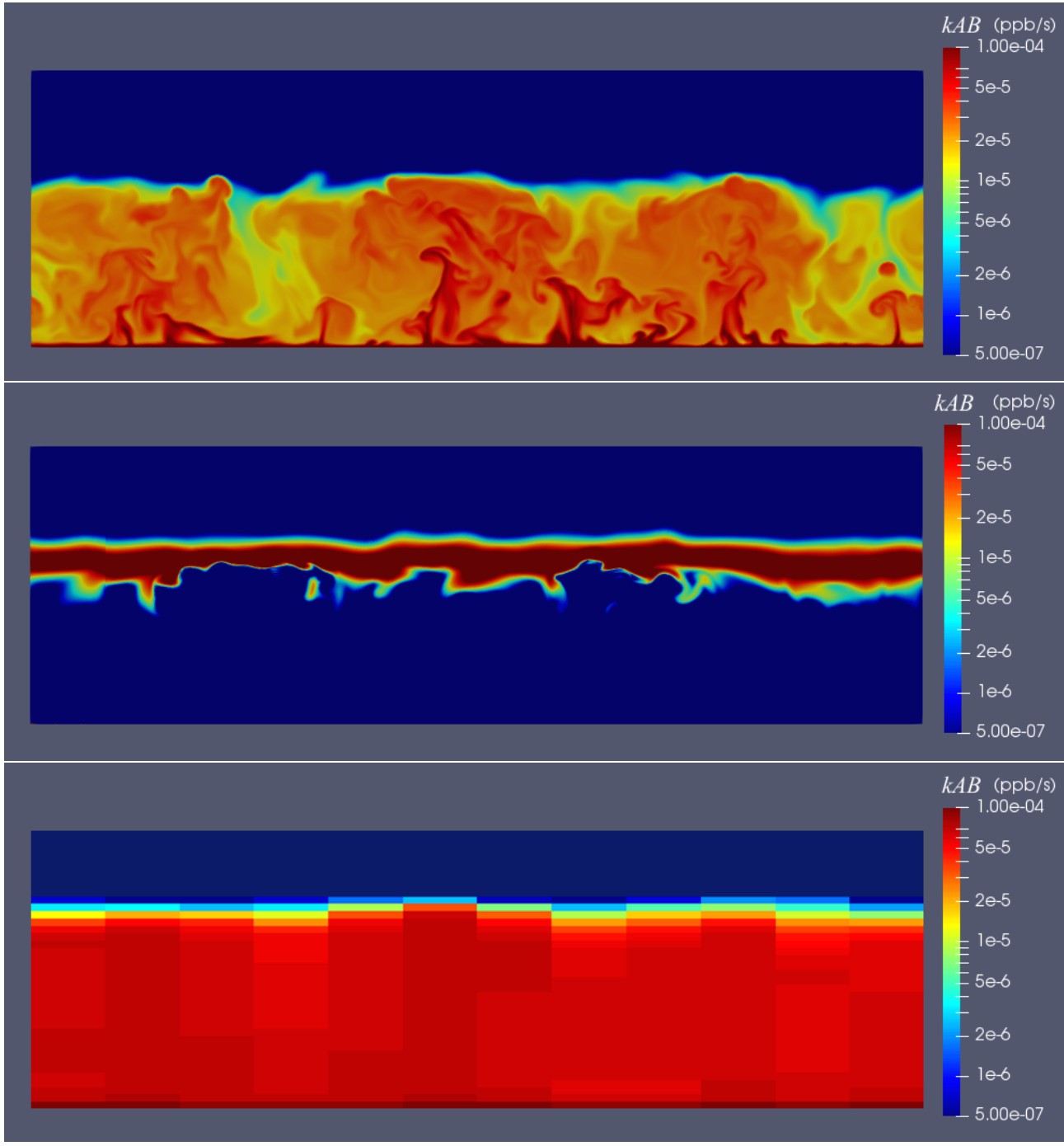

**Figure 2.** Colour maps of the distribution of the production term ($kAB$) at the end of the simulations with homogeneous emissions for the cases slow-VV05 (*top panel*) and slow-mflux (*middle panel*). *Bottom panel*: Same colour map for the same case as in the top panel, but in a 1 km-64 lev coarse-grid model.



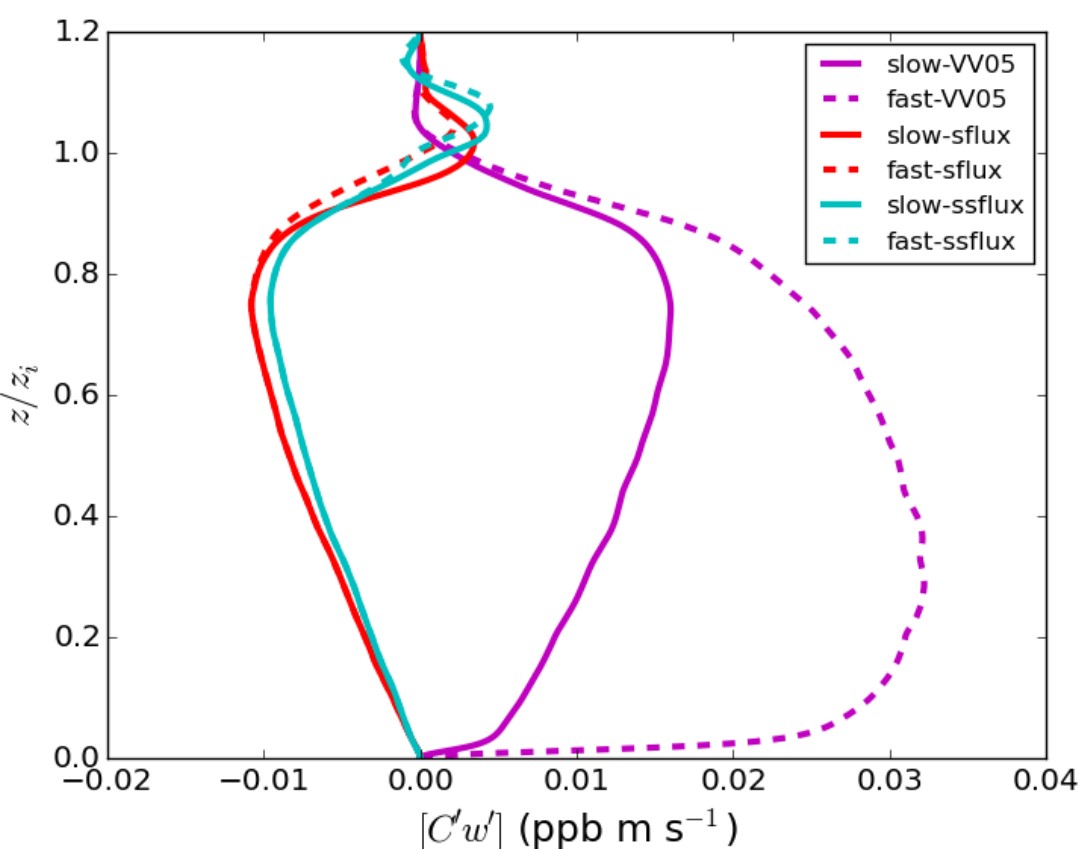

**Figure 3.** Vertical profiles of the horizontally-averaged vertical flux of Tracer C $[C'w']$ with homogeneous emissions for varied cases time-averaged for the last 30 minutes of the simulations.

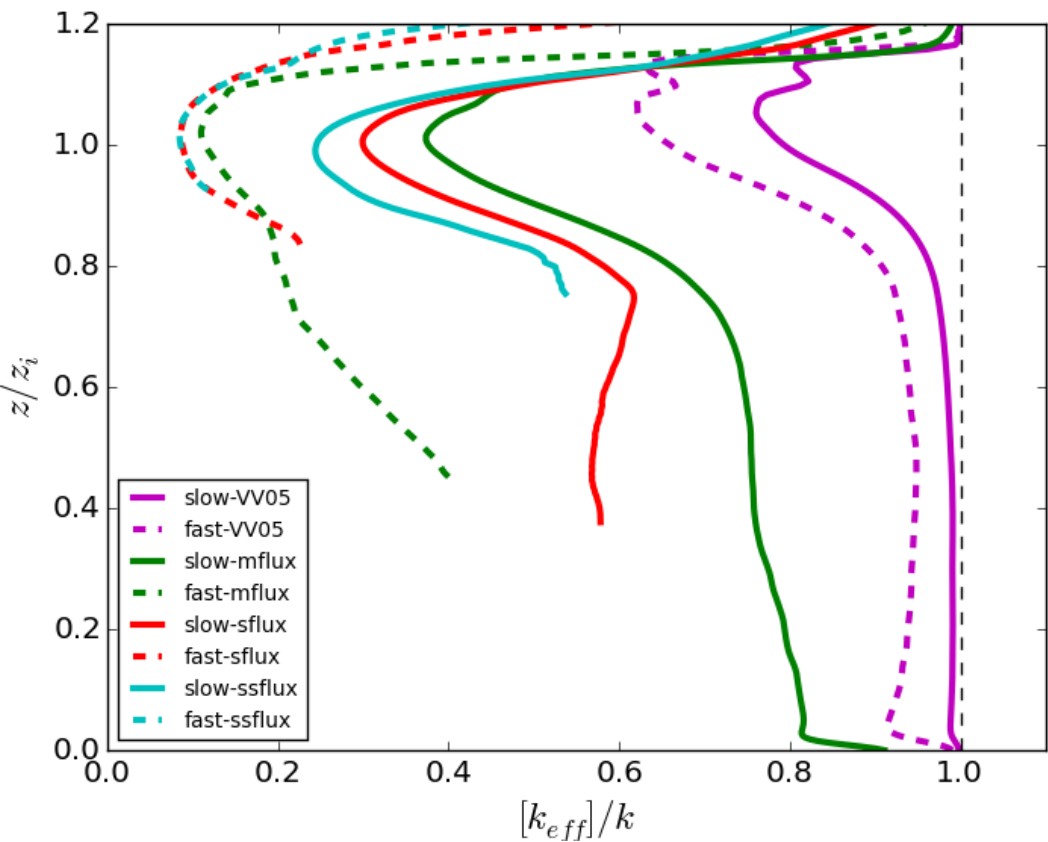

**Figure 4.** Vertical profiles of the horizontally-averaged normalised effective chemical reaction rate $[k_{eff}]/k$ with homogeneous emissions for varied cases time-averaged for the last 30 minutes of the simulations. Note that the vertical profiles are not plotted at altitudes where the concentration of Tracer B is equal to zero.

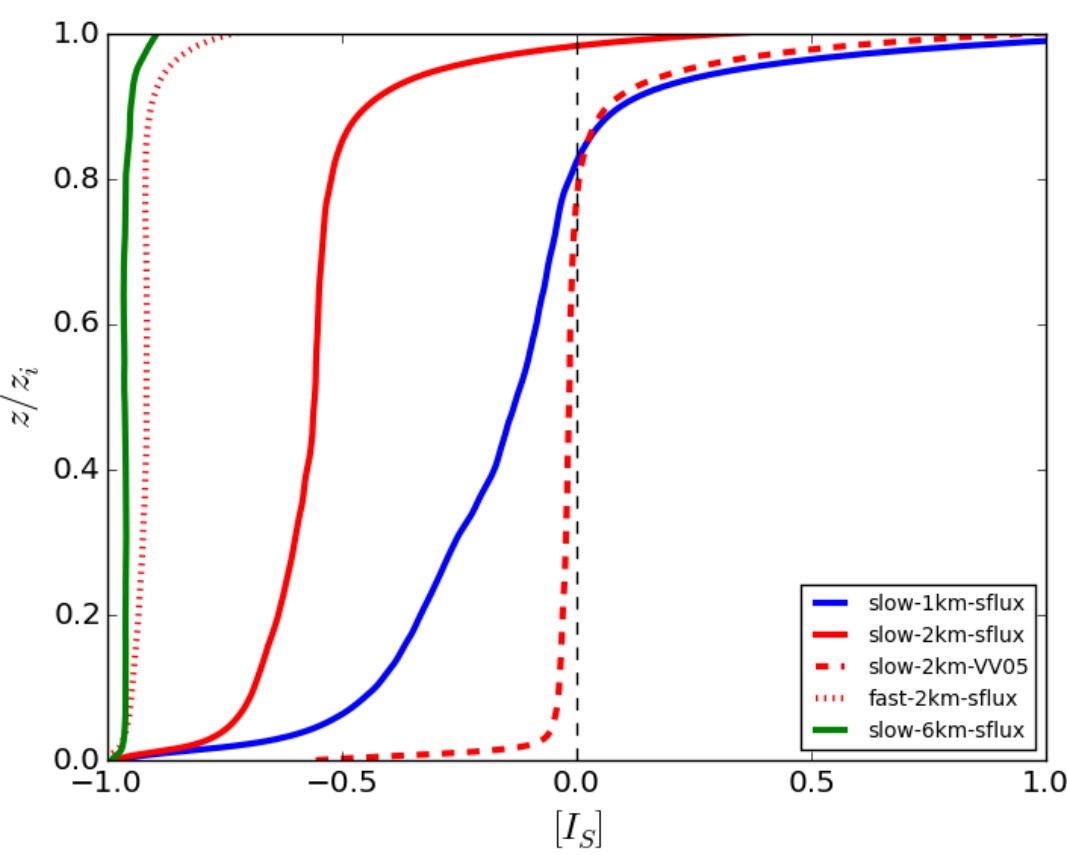

**Figure 5.** Vertical profiles of the horizontally-averaged segregation coefficient $[I_S]$ with heterogeneous emissions for varied cases time-averaged for the last 30 minutes of the simulations.



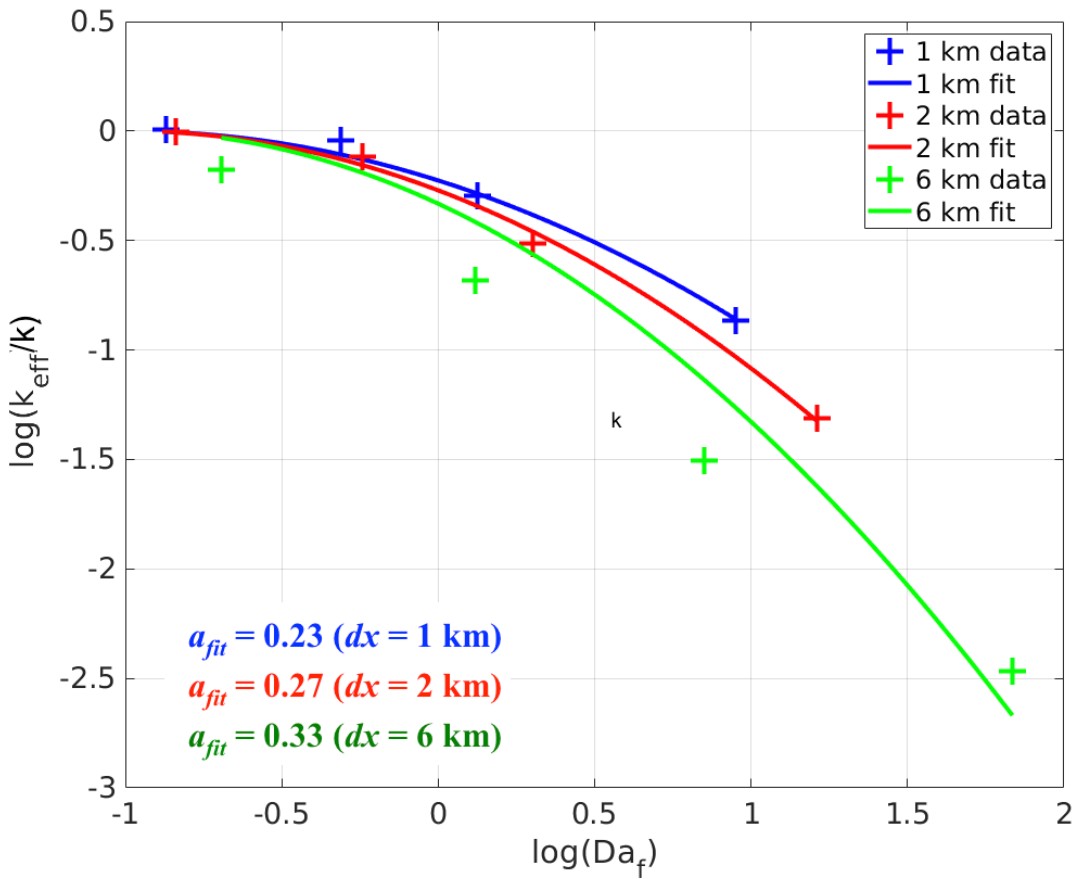

**Figure 6.** Fitted curves of the normalised boundary layer-averaged effective chemical reaction rate $k_{eff}/k$ against the final Damköhler numbers $Da_f$ for the simulations with heterogeneous emissions with different length of heterogeneity $(dx)$. $a_{fit}$ refers to the fitted slope for the square law between $\log(k_{eff}/k)$ and $\log(Da_f)$ corresponding to different $dx$.

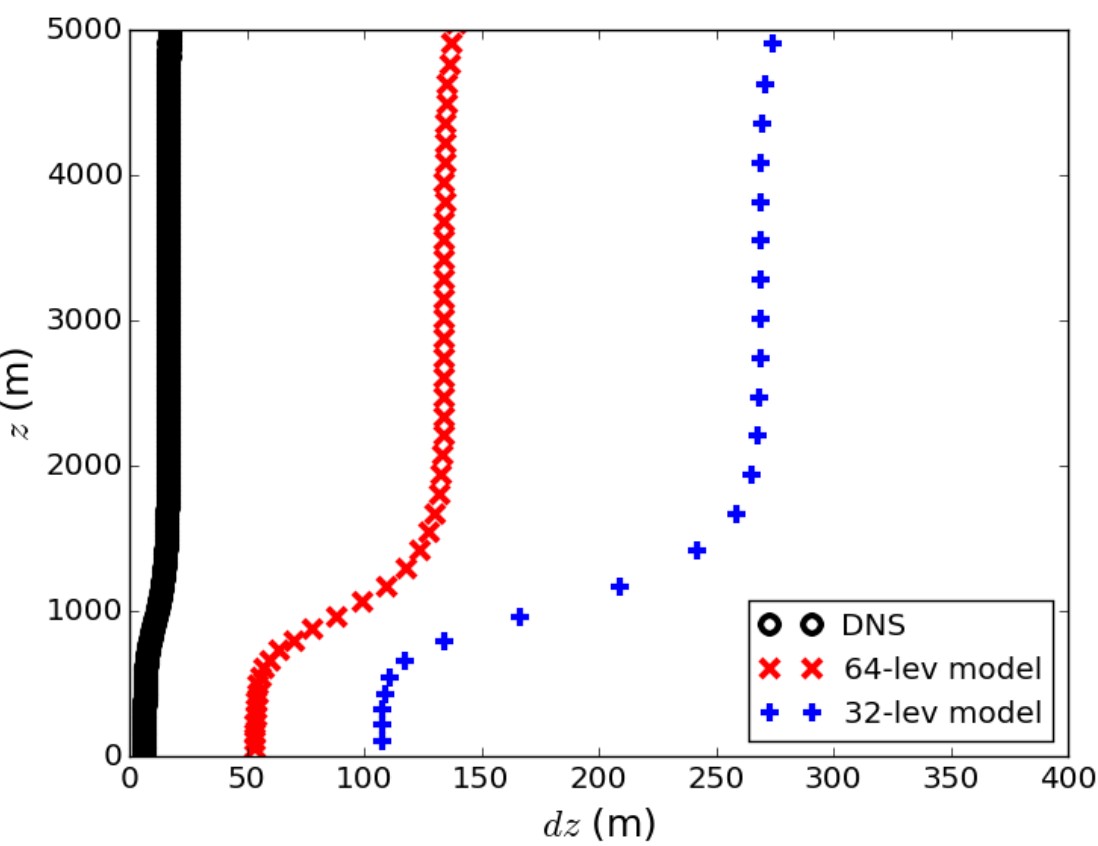

**Figure 7.** Plot of the vertical grid spacing ($dz$) (in equivalent to vertical resolution) versus height ($z$) in the DNS model (*black*), the 64-lev coarse-grid models (*red*) and the 32-lev coarse-grid models (*blue*) in the first 5 km.



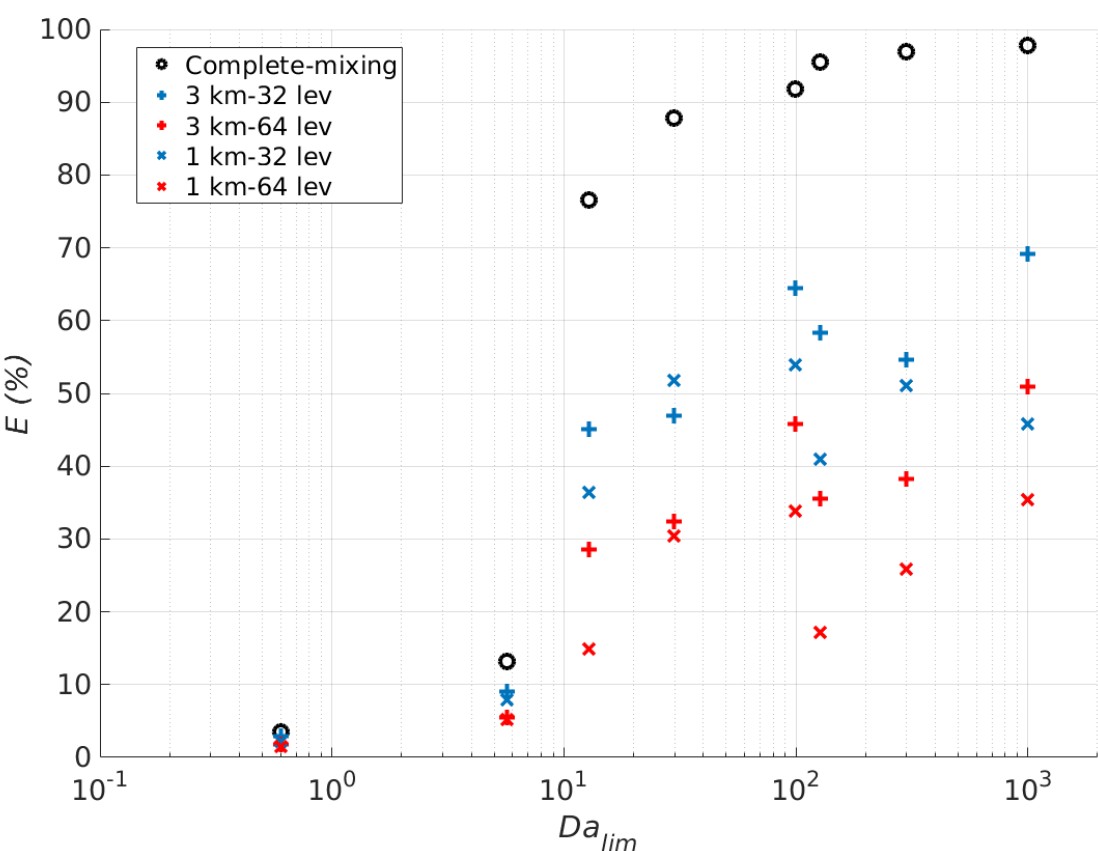

**Figure 8.** Plot of the model errors $E$ in percentage from the complete-mixing model (black circles) and the four coarse-grid models (+ and × in red and blue) against the corresponding Damköhler number of the limiting reactant ($Da_{lim}$) for the simulations with homogeneous emissions.



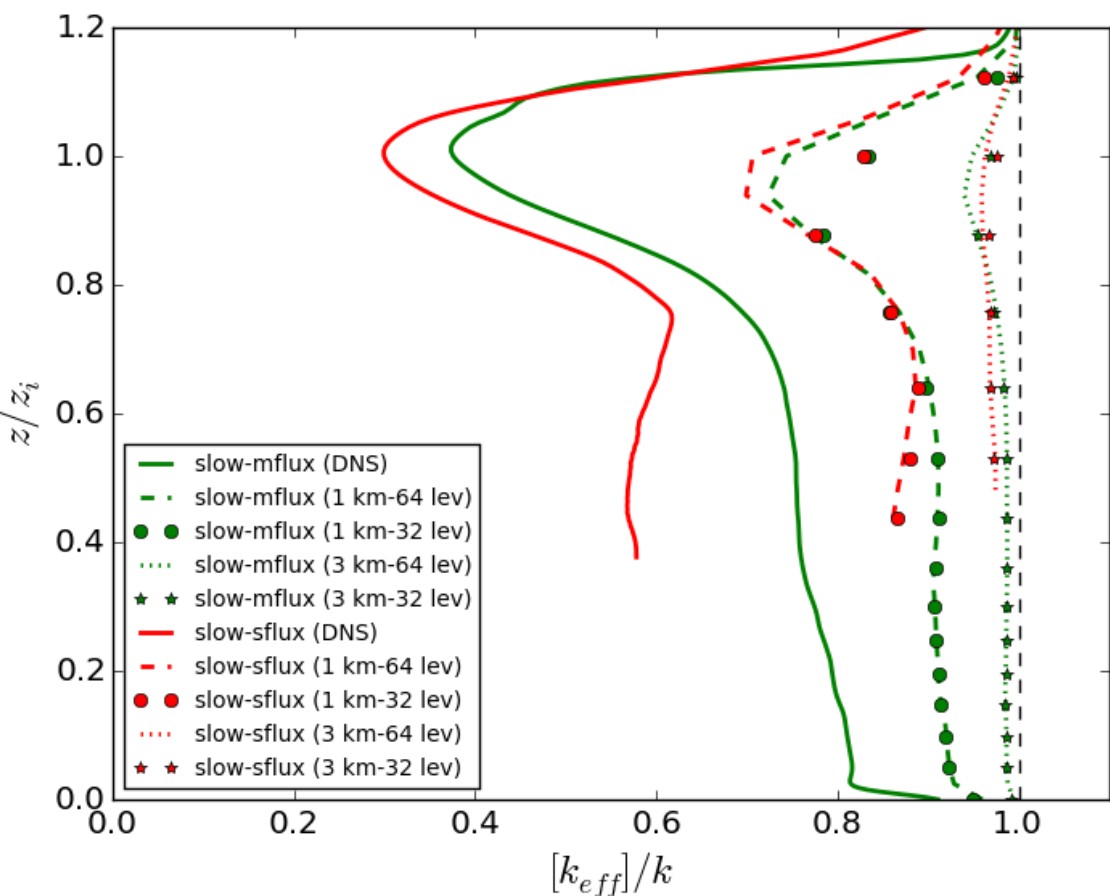

**Figure 9.** Vertical profiles of the horizontally-averaged normalised effective chemical reaction rate $[k_{eff}]/k$ from the four coarse-grid models with homogeneous emissions for the slow-mflux and slow-sflux cases time-averaged for the last 30 minutes of the simulations. Note that the vertical profiles are not plotted at altitudes where the concentration of Tracer B is equal to zero.



**Figure 10.** Plot of model errors in percentage, categorised with different length of heterogeneity ($dx$), from the four coarse-grid models ($E$) against the corresponding errors from the complete-mixing model ($E_{cm}$) for the simulations with heterogeneous emissions.





| (a) Homogeneous emissions | | |
|---|---|---|
| Case | $k$ (ppb$^{-1}$ s$^{-1}$) | $F_A$ (ppb m s$^{-1}$) |
| slow-VV05 | $4.75 \times 10^{-4}$ | 0.05 |
| fast-VV05 | $4.75 \times 10^{-3}$ | 0.05 |
| slow-mflux | $4.75 \times 10^{-4}$ | 0.25 |
| fast-mflux | $4.75 \times 10^{-3}$ | 0.25 |
| slow-sflux | $4.75 \times 10^{-4}$ | 0.50 |
| fast-sflux | $4.75 \times 10^{-3}$ | 0.50 |
| slow-ssflux | $4.75 \times 10^{-4}$ | 1.41 |
| fast-ssflux | $4.75 \times 10^{-3}$ | 1.41 |

| (b) Heterogeneous emissions | | | |
|---|---|---|---|
| Case | $dx$ (km) | $k$ (ppb$^{-1}$ s$^{-1}$) | $F_A$ (ppb m s$^{-1}$) |
| $dx = 1$ km | | | |
| slow-1km-VV05 | 1 | $4.75 \times 10^{-4}$ | 0.05 |
| fast-1km-VV05 | 1 | $4.75 \times 10^{-3}$ | 0.05 |
| slow-1km-sflux | 1 | $4.75 \times 10^{-4}$ | 0.5 |
| fast-1km-sflux | 1 | $4.75 \times 10^{-3}$ | 0.5 |
| $dx = 2$ km | | | |
| slow-2km-VV05 | 2 | $4.75 \times 10^{-4}$ | 0.05 |
| fast-2km-VV05 | 2 | $4.75 \times 10^{-3}$ | 0.05 |
| slow-2km-sflux | 2 | $4.75 \times 10^{-4}$ | 0.5 |
| fast-2km-sflux | 2 | $4.75 \times 10^{-3}$ | 0.5 |
| $dx = 6$ km | | | |
| slow-6km-VV05 | 6 | $4.75 \times 10^{-4}$ | 0.05 |
| fast-6km-VV05 | 6 | $4.75 \times 10^{-3}$ | 0.05 |
| slow-6km-sflux | 6 | $4.75 \times 10^{-4}$ | 0.5 |
| fast-6km-sflux | 6 | $4.75 \times 10^{-3}$ | 0.5 |

**Table 1.** The names and simulation parameters of the DNS simulations, including (a) the imposed chemical reaction rate $k$ and the surface emission flux of Tracer A $F_A$ for the simulations with homogeneous emissions, (b) plus the length of heterogeneity ($dx$) for the simulations with heterogeneous emissions.





| (a) Homogeneous emissions | | | | | |
|---|---|---|---|---|---|
| Case | $Da_{A,i}$ | $Da_{B,i}$ | $Da_{A,f}$ | $Da_{B,f}$ | $k_{eff}/k$ |
| slow-VV05 | 0.1438 | 0.0094 | 0.6051 | 0.0405 | 0.9652 |
| fast-VV05 | 1.4377 | 0.0938 | 5.7148 | 0.0688 | 0.8687 |
| slow-mflux | 0.1531 | 0.0291 | 0.0143 | 12.7801 | 0.2350 |
| fast-mflux | 1.5306 | 0.2907 | 0.0472 | 127.6096 | 0.0455 |
| slow-sflux | 0.1531 | 0.0214 | 0.0028 | 29.8211 | 0.1225 |
| fast-sflux | 1.5306 | 0.2142 | 0.0033 | 298.1865 | 0.0304 |
| slow-ssflux | 0.1531 | 0.0560 | 0.0002 | 99.7068 | 0.0819 |
| fast-ssflux | 1.5306 | 0.5602 | $4.8058 \times 10^{-5}$ | 997.0659 | 0.0226 |
| (b) Heterogeneous emissions | | | | | |
| Case | $Da_{A,i}$ | $Da_{B,i}$ | $Da_{A,f}$ | $Da_{B,f}$ | $k_{eff}/k$ |
| $dx = 1$ km | | | | | |
| slow-1km-VV05 | 0.0029 | 0.0029 | 0.1347 | 0.1347 | 1.0184 |
| fast-1km-VV05 | 0.0290 | 0.0289 | 0.4896 | 0.4894 | 0.9094 |
| slow-1km-sflux | 0.0070 | 0.0070 | 1.3364 | 1.3392 | 0.5053 |
| fast-1km-sflux | 0.0740 | 0.0740 | 8.9180 | 8.9452 | 0.1365 |
| $dx = 2$ km | | | | | |
| slow-2km-VV05 | 0.0024 | 0.0024 | 0.1442 | 0.1453 | 0.9952 |
| fast-2km-VV05 | 0.0241 | 0.0241 | 0.5656 | 0.5766 | 0.7634 |
| slow-2km-sflux | 0.0069 | 0.0069 | 1.9957 | 2.0217 | 0.3068 |
| fast-2km-sflux | 0.0686 | 0.0686 | 16.1687 | 16.4241 | 0.0487 |
| $dx = 6$ km | | | | | |
| slow-6km-VV05 | 0.0036 | 0.0036 | 0.2017 | 0.2034 | 0.6650 |
| fast-6km-VV05 | 0.0357 | 0.0357 | 1.3034 | 1.3213 | 0.2068 |
| slow-6km-sflux | 0.0110 | 0.0110 | 7.0356 | 7.1541 | 0.0312 |
| fast-6km-sflux | 0.1070 | 0.1070 | 67.8306 | 69.0153 | 0.0034 |

**Table 2.** The initial Damköhler numbers $Da_{A,i}$ and $Da_{B,i}$, and final Damköhler numbers $Da_{A,f}$ and $Da_{B,f}$ with the resultant normalised effective chemical reaction rate $k_{eff}/k$ in the DNS simulations with (a) homogeneous emissions and (b) heterogeneous emissions.





| (a) Homogeneous emissions | | | | | |
|---|---|---|---|---|---|
| Case | Complete-mixing | 1 km-64 lev | 1 km-32 lev | 3 km-64 lev | 3 km-32 lev |
| slow-VV05 | 3.48 | 1.43 | 2.18 | 1.73 | 2.94 |
| fast-VV05 | 13.13 | 5.26 | 7.97 | 5.44 | 8.98 |
| slow-mflux | 76.50 | 14.82 | 36.41 | 28.57 | 45.15 |
| fast-mflux | 95.45 | 17.16 | 40.94 | 35.57 | 58.29 |
| slow-sflux | 87.75 | 30.39 | 51.84 | 32.40 | 46.92 |
| fast-sflux | 96.96 | 25.87 | 51.09 | 38.31 | 54.62 |
| slow-ssflux | 91.81 | 33.84 | 53.95 | 45.79 | 64.48 |
| fast-ssflux | 97.74 | 35.42 | 45.83 | 50.97 | 69.14 |
| (b) Heterogeneous emissions | | | | | |
| Case | Complete-mixing | 1 km-64 lev | 1 km-32 lev | 3 km-64 lev | 3 km-32 lev |
| $dx = 1$ km | | | | | |
| slow-1km-VV05 | -1.84 | 3.58 | 1.50 | 3.13 | 1.27 |
| fast-1km-VV05 | 9.06 | 38.36 | 27.95 | 38.6 | 28.34 |
| slow-1km-sflux | 49.47 | 128.00 | 100.64 | 130.45 | 102.49 |
| fast-1km-sflux | 86.35 | 266.47 | 203.62 | 274.01 | 209.51 |
| $dx = 2$ km | | | | | |
| slow-2km-VV05 | 0.48 | -0.17 | -1.22 | 4.39 | 2.82 |
| fast-2km-VV05 | 23.66 | 7.76 | 7.08 | 40.13 | 32.07 |
| slow-2km-sflux | 69.32 | 29.66 | 30.24 | 121.83 | 102.08 |
| fast-2km-sflux | 95.13 | 35.95 | 37.01 | 176.45 | 145.97 |
| $dx = 6$ km | | | | | |
| slow-6km-VV05 | 33.50 | 7.34 | 6.96 | 12.93 | 12.54 |
| fast-6km-VV05 | 79.32 | 12.63 | 12.67 | 25.57 | 25.47 |
| slow-6km-sflux | 96.88 | 8.18 | 8.30 | 18.55 | 18.48 |
| fast-6km-sflux | 99.66 | 6.88 | 7.02 | 16.41 | 16.39 |

**Table 3.** Resultant model errors in percentage from the complete-mixing model ($E_{cm}$) and the four coarse-grid models ($E$) for the simulations with (a) homogeneous emissions and (b) heterogeneous emissions.