# Peer review of "Error induced by neglecting subgrid chemical segregation due to inefficient turbulent mixing in regional chemical-transport models in urban environments"

_Atmospheric Chemistry and Physics, 2020_

## Referee Comment (RC1) · Jordi Vila-Guerau de Arellano (Referee) · 3 Aug 2020

This research presents very interesting results on a complex and interdisciplinary topic that is not yet solved, how the turbulence influence chemistry over (urban) heterogeneous emissions. The paper confirms previous results to take into account and the need to parameterize the effects of inhomogeneous mixing. The originality of the work is to quantify the errors due to the segregation of species in air quality models and its dependence on the vertical and horizontal resolutions. In that respect, I think the use of direct numerical simulation is very welcomed in the field and the reader of ACP will

appreciate this sort of research. I have written my comments section by section.

**1. Introduction**

I found the introduction very complete and very well-written. The aim of the paper, its research strategy and the methods are very well posed and explained. A potential improvement to gain clarity is to make a clearer distinction to the segregation of chemical species due to the state of nature of turbulence and their drivers - thermal stability, canopy-atmosphere interactions, clouds - and the one due to the heterogeneity of the surface emissions. I realize that they are closely related, but in my opinion, the reader will appreciate better the complexity of the problem if this distinction is made. From the very complete and literature review that shows the capacity of synthesis and the deep understanding of the topic by the authors, I miss the following papers: Patton, E. G., et al. (2001): Decaying scalars emitted by a forest canopy - A numerical study, Boundary-Layer Meteorology, 100, pp. 91-12 and Baker J. (2004): A study of the dispersion and transport of reactive pollutants in and above street canyons—a large eddy simulation. Atmospheric Environment 38, 6883-6892.

**2. Model description**

I would like to congratulate the authors of using the direct numerical simulation technique to move forward in the complex topic of the interactions between turbulence and chemistry. This numerical technique sheds new light, and it is complementary to the large-eddy simulation technique. I have in this section the following comments:

a) The authors defined the Damkohler number as a function of the most energetic turbulent time scale. However, I was wondering on the need to define a Damkohler number to quantify the effects of turbulence at the smaller spatiotemporal scales, for instance using a Kolmogorov time scale. Chemical reactions occur at the molecular diffusion time scale, and I wonder whether it is necessary to include a physical time scale closer to this one of chemistry. I think the reader would like to hear the opinion of the authors in that respect and if their DNS numerical experiments could add extra

information.

b) At section 2.2.1, the authors focus on the most extreme case of segregation, i.e. an irreversible second-order reaction. I understand their choice, but I think the reader will appreciate if they could include in their chemical mechanism a backward reaction. By so doing, the chemical will become closer to an equilibrium, and it will probably reduce the effects of the segregation. In that respect, the analytical solution for three chemical species and two reactions derived by Jonker et al. (2004) (Characteristics length scales of reactive species in the convective boundary layer, Journal of Atmospheric Sciences 61, 41-55) can be very useful to study the effect without introducing additional computational costs. Note that variations on time and on space due to presence of clouds or vertical perturbances of the radiation transfer within the urban canopy can be interested cases to bring this triad of species out of equilibrium, and therefore enhance the segregation effect again.

3. Results

- Homogeneous emission. For the sake of completeness, I think it might be interesting in Figure 2 to add an extra panel but now with a typical time average used by the air quality models ($\sim$30 minutes) to (probably) show how the small-scale fluctuations of the chemical term are also filtered out due to the time-averaging.

- Section 3.1.2. (lines 12-14). I think the authors are reporting a very interesting results regarding the more prominent minima due to the higher vertical resolution. Could they please elaborate and quantified a bit more?

- Since the paper is entitled "errors", I think an additional figure 3 that can complete the discussion is the calculation of the exchange coefficient K defined as the ratio between the turbulent flux and the mean gradient). I realize that in the middle of the CBL can be undetermined, but close to the surface and at the entrainment zone it might provide information on the combined effect of chemistry and turbulence on the parameterization of the transport of chemical species.

- Section 3.2 (line 29). It might be interesting to show how this Is = - 0.8 changes if a first-order backward reaction is included in the chemical mechanism of reaction R1.

- Figure 9. Could the authors elaborate more the reasons of lower values on k_eff/k at the entrainment zone compared to the values near the surface? Both regions are characterized by higher gradients of the chemically active species. Why are these differences at these two regions?

4. Discussion

I think it can be worth to make a discussion point that the errors are not only on the segregation and therefore on the chemical reaction rate, but also on the eddy diffusivity and therefore on the transport of chemical species.

Section 5.1. (lines 9-16) Will look-up tables depending on the processes that they hace mentioned be a solution instead of the parameterizations?

Page 17 and line 1. Will previous work on natural canopies be useful to be connected to urban canopies?

Since section 5.2 presents an outlook of possibilities for future research, I think it might important to include a short description on the effects of radiation and its perturbation (clouds, urban canopy) closely couple to the turbulent effect. An additional experiment including the backward reaction as here proposed can pave the way to future research.

---

## Referee Comment (RC2) · Anonymous Referee #3 · 20 Nov 2020

Error induced by neglecting subgrid chemical segregation due to inefficient turbulent mixing in regional chemical-transport models in urban environments

General comments This study investigates and quantifies the impacts of chemical segregation on chemical reaction simulation which is a highly relevant topic in the field of turbulence-chemistry interaction. The DNS is an appropriate method for this study. The experiments are well designed regarding the strong and heterogeneous emissions. The results are very well organized and interpreted, and implications of the results are properly discussed. A couple of my concerns are appropriately acknowledged or dis-

cussed such as the potential influences of multiple chemical reactions, the modeling framework of regional models, and different weather conditions. I only have a few minor comments mostly regarding clarification. Please see below.

Specific comments 1. Figure 2. Can you note the x axis in this figure? 2. Figure 4&5. Is there a particular reason you use Is in Fig. 5 and keff/k in Fig. 4? Could you keep it consistent? 3. Figure 9. I wonder if you run the simulations with the fast chemistry. If so, how different are they from the slow chemistry cases shown here? 4. Equation 1. Please define Fb. 5. Page 8 Line 12. Add the acronym CBL to where the full name first appears. 6. Page 13 Line 4. Can you clarify "interpolated"?

---

## Author Comment (AC2) · 25 Nov 2020

We first would like to thank you for your positive comments to our manuscript. It is good to learn that our manuscript has acknowledged most of your main concerns on the topic.

Regarding your specific comments, we would like to address them point-by-point:

(1) The *x*-axis refers to the horizontal dimension of the simulation domain. We will add the label of the *x*-axis to indicate that in the next revision of the manuscript as in the following figure:

[Figure]

(2) The segregation coefficient $I_S$ is presented in Figure 5 to match with the description in Section 3.2. We think that it will be easier to explain that the flows of Tracer A and B are co-related in the situation with $I_S > 0$ with the expression of $I_S$ instead of $k_{eff}$. We suggest to keep the *x*-axis in $I_S$ but to add an additional *x*-axis in $k_{eff}$ in Figure 5 for consistency as the below figure:

[Figure]

(3) We have also performed similar runs for the fast chemistry cases in Figure 9. With fast chemistry, the underestimation of $[E]$ would be more significant, but the locations with the most prominent underestimation are still at the top of the surface layer and at the top of the boundary layer. We will present the fast chemistry case with the mflux emission flux scenario in Figure 9 for a comparison in the next revision of the manuscript as in the following figure:

[Figure]

(4) $F_b$ is in fact defined on Page 5 Line 8.

(5) Thanks for the reminder. We will add the definition of the acronym CBL when the term "convective boundary layer" first appears on Page 2 Line 21:

> Many of these LES studies focus on the convective boundary layer ==(CBL),== in which the imbalance between updraft and downdraft transport produces a large segregation of the reactants (Wyngaard and Brost, 1984; Chatfield and A. Brost, 1987).

(6) The high-resolution grids of the DNS simulations are mapped onto the lower-resolution grids of the coarse-grid models. The concentration fields obtained from the DNS simulations are then interpolated from the high-resolution DNS model grids to the lower-resolution coarse-grid model grids using local-area averaging. We have rephrased the description of the coarse-grid model method in Page 13 Line 4-6:

> . ==The tracer concentration fields obtained from the DNS simulations are interpolated from the high-resolution DNS model grids to the lower-resolution coarse-grid model grids.== The volumetric averages of tracer concentrations in each model grid are then calculated. The statistics of these resolution-degraded concentration fields are then calculated as in Section 3.

We truly hope you agree that these revisions can improve the presentation of the manuscript.

---

## Author Response (AR1)

**Author's Response for the manuscript** *Error induced by neglecting subgrid chemical segregation due to inefficient turbulent mixing in regional chemical-transport models in urban environments* (acp-2020-545)

5     The reviews and the responses to the reviews from both Reviewer #1 (Jordi Vila-Guerau de Arellano) and Anonymous Referee #3 are included and presented point-by-point in this document. The marked-up manuscript with the revision is attached at the end of the document.

**(I) Reply to Reviewer #1 (Jordi Vila-Guerau de Arellano)**

**Review #1:**

This research presents very interesting results on a complex and interdisciplinary topic that is not yet solved, how the turbulence influence chemistry over (urban) heterogeneous emissions. The paper confirms previous results to take into account and the need to parameterize the effects of

15     inhomogeneous mixing. The originality of the work is to quantify the errors due to the segregation of species in air quality models and its dependence on the vertical and horizontal resolutions. In that respect, I think the use of direct numerical simulation is very welcomed in the field and the reader of ACP will appreciate this sort of research. I have written my comments section by section.

20     **Response #1:**

We would like to first thank you for your very kind and constructive comments. We would like to reply your comments point by point in sections.

**1. Introduction**

**Review #2:**

30     I found the introduction very complete and very well-written. The aim of the paper, its research strategy and the methods are very well posed and explained. A potential improvement to gain clarity is to make a clearer distinction to the segregation of chemical species due to the state of nature of turbulence and their drivers - thermal stability, canopy-atmosphere interactions, clouds - and the one due to the heterogeneity of the surface emissions. I realize that they are closely related,

35     but in my opinion, the reader will appreciate better the complexity of the problem if this distinction is made. From the very complete and literature review that shows the capacity of synthesis and the deep understanding of the topic by the authors, I miss the following papers: Patton, E. G., et al. (2001): Decaying scalars emitted by a forest canopy - A numerical study, Boundary-Layer Meteorology, 100, pp. 91-12 and Baker J. (2004): A study of the dispersion and transport of

40     reactive pollutants in and above street canyons - a large eddy simulation. Atmospheric Environment 38, 6883-6892.

**Response #2:**

45     Thank you very much for your very positive comments on the introduction, and your suggestion for additional points and references to make the introduction more complete. We have added the following highlighted sentence in the first paragraph of the introduction to address your point about the two origins of chemical segregation:

50     Turbulence mixes initially-segregated reactive species in the boundary layer, and allows chemical reactions to occur. However, for fast chemical reactions with the chemical timescale shorter than the turbulent timescale, turbulent motions mix the reactants so slowly that they remain segregated rather than reacting. ==This segregation can be a result of the inefficient mixing due to the state of turbulence and its driver, such as thermal stabiliy, canopy-atmosphere interaction and cloud processes, and/or a result of the heterogeneity of surface emissions (Vilà-Guerau==

55     ==de Arellano, 2003).==

The two references you have suggested are also added in the introduction (see highlighted text in the introduction paragraphs):

Many of these LES studies focus on the convective boundary layer, in which the imbalance between updraft and downdraft transport produces a large segregation of the reactants (Wyngaard and Brost, 1984; Chatfield and A. Brost, 1987). Such LES studies were often performed with idealised cases with a bottom-up tracer emitted from the surface and top-down tracer entrained from the free troposphere with a simple second-order chemistry scheme. For instance, Schumann (1989) pointed out that the segregation between the two tracers depends on the Damköhler number ($Da$), the concentration ratio of the two species and the specified initial conditions, pinning the use of $Da$ as an indicator to estimate whether turbulent motions are significantly affecting chemical reactions in the flow. Vinuesa and Vilà-Guerau de Arellano (2005) introduced the concept of an effective chemical reaction rate ($k_{eff}$) to quantify the actual boundary layer-averaged reaction rate that accounts for the effect of the chemical segregation. They also reported a drop of $k_{eff}$ up to 20% from the imposed chemical reaction rate $k$ when $Da \sim 1$, while $k_{eff} \sim k$ when $Da \sim 0.1$. Recently, the effect of segregation due to inefficient turbulent mixing on chemical reaction has been considered as the cause of the miscalculation of large-scale models in a number of scenarios. One of the most discussed issues is the under-prediction of the concentration of hydroxyl radical (OH) in global models. A number of studies employed LES models over forestal areas with sophisticated chemical mechanisms involving biogenic VOCs to simulate the resultant segregation between isoprene and OH (e. g. Patton et al. (2001); Brosse et al. (2017); Dlugi et al. (2019)). However, the magnitude of the segregation coefficient $I_S$ shown in these studies is in general less than 20%, which is too small to explain the observed discrepancy, where $I_S \sim 50\%$ is necessary (Butler et al., 2008).

The aim of the present work is to investigate the effect of inefficient turbulent mixing on chemical reactions in an urban-like boundary layer with strong and heterogeneously-distributed surface emissions, and to account for the errors induced by neglecting the resultant subgrid chemical segregation in relatively coarse regional models. While previous studies focused on agricultural and rural conditions where the emission fluxes are relatively low ($\sim O(0.01)$ ppb m s$^{-1}$), our simulations address cases of strong emission fluxes in typical urban values ($\sim O(0.1$-$1.0)$ ppb m s$^{-1}$). In two rare related studies in urban air condition, Baker et al. (2004) reported from their LES model of an urban street canyon significant deviations of the concentrations of the NO-NO2-O3 triad from the equilibrated values to the photostationary state depending on the turbulent structure in the canyon; while Auger and Legras (2007) obtained high values of instantaneous segregation under certain emission configurations in urban areas with a chemical system of 44 species. On top of their conclusions, our study aims to explain why this strong segregation occurs under urban conditions and on which parameters the errors induced by neglecting the segregation in large-scale models depend. To achieve this aim, we perform DNS simulations with homogeneous emissions with an idealised second-order A-B-C chemistry scheme (A+B➔C) with emission fluxes extended to urban values, in addition to a set of simulations with heterogeneous emissions. With varied reaction rates, the idealised second-order A-B-C chemistry scheme can generally represent any second-order chemical reactions commonly seen in an urban environment.
* * *
**2. Model description**

**Review #3:**

I would like to congratulate the authors of using the direct numerical simulation technique to move forward in the complex topic of the interactions between turbulence and chemistry. This numerical technique sheds new light, and it is complementary to the large-eddy simulation technique. I have in this section the following comments:

**Response #3:**

It is a pleasure to hear that you find the direct numerical simulation technique a plus to the research of the topic.
* * *
**Review #4:**

a) The authors defined the Damkohler number as a function of the most energetic turbulent time scale. However, I was wondering on the need to define a Damkohler number to quantify the effects of turbulence at the smaller spatiotemporal scales, for instance using a Kolmogorov time scale. Chemical reactions occur at the molecular diffusion time scale, and I wonder whether it is necessary to include a physical timescale closer to this one of chemistry. I think the reader would like to hear the opinion of the authors in that respect and if their DNS numerical experiments could add extra information.

**Response #4:**

(a) The Kolmogorov timescale is around 10 s for all the simulations. We adopt the definition of the corresponding Kolmogorov Damköhler number in Vilà-Guerau de Arellano et al. (2004). The definition of the corresponding Kolmogorov Damköhler number is added in the manuscript in Section 2.3 after the description of the Damköhler number:

> To quantify the chemical-turbulence interaction at the molecular diffusion spatiotemporal scale, the Kolmogorov Damköhler numbers are also calculated in some of simulations. The definition of Kolmogorov Damköhler number is adopted from Vilà-Guerau de Arellano et al. (2004), i.e.:
>
> $$Da_{k,A} = \frac{t_k}{t_{chem,A}}$$
>
> for Tracer A, and similarly for Tracer B ($Da_{k,B}$) by replacing the denominator with $t_{chem,B}$. The Kolmogorov timescale $t_k$ is around 10 s for all the simulations (Garcia and Mellado, 2014).

Since the dynamical settings of the DNS simulations in the paper are the same (except for the longer simulation time for the heterogeneous runs), the Kolmogorov Damköhler numbers of the runs are equal to the Damköhler numbers provided in Table 2 in the original draft divided by the ratio between the convective and Kolmogorov timescale (around 79.8). The resultant Kolmogorov Damköhler numbers are given in Table 1 of this document. For comparison, we have replotted Figure 8 of the original draft to include an additional *x*-axis with the limiting Kolmogorov Damköhler number ($Da_{k,lim}$) (see Figure 1 of this document). The errors induced by neglecting the subgrid chemical segregation become significant when $Da_{k,lim}$ reaches the order of around $10^{-1}$. We will replace Figure 8 of the original manuscript with Figure 1 of this document, and to add the following highlighted text addressing this issue in Section 3.13:

> Since the chemical transformation of the reactant that is relatively less abundant than the other reactant is influenced more by turbulent mixing (Vilà-Guerau de Arellano et al., 2004), $Da_{B,f}$ is now a better indicator for the role of turbulent mixing on chemical reactions than $Da_{A,f}$. Summarising the simulations with homogeneous emissions, one can observe from Figure 8 (black circles) that the deviation of $k_{eff}$ from the imposed rate $k$, or the error from the complete-mixing model, increases with the increased Damköhler number of the limiting reactant ($Da_{lim}$), where $Da_{lim} = Da_{A,f}$ when the reaction is Tracer A-limiting and $Da_{lim} = Da_{B,f}$ when the reaction is Tracer B-limiting. This transition occurs when $Da_{lim}$ reaches the order of 10. Similar concept can also applied to the corresponding Kolmogorov Damköhler number, where the limiting Kolmogorov Damköhler number ($Da_{k,lim}$) is introduced (refer to the upper *x*-axis of Figure 8). The errors induced by neglecting the subgrid chemical segregation become significant when $Da_{k,lim}$ reaches the order of around $10^{-1}$.

| (a) Homogeneous emissions | | | | | |
|---|---|---|---|---|---|
| Case | $Da_{k\,A,i}$ | $Da_{k\,B,i}$ | $Da_{k\,A,f}$ | $Da_{k\,B,f}$ | $k_{eff}/k$ |
| slow-VV05 | 7.25E-03 | 4.73E-04 | 8.44E-03 | 5.65E-04 | 0.9652 |
| fast-VV05 | 7.25E-02 | 4.73E-03 | 7.97E-02 | 9.60E-04 | 0.8687 |
| slow-mflux | 7.72E-03 | 1.47E-03 | 1.99E-04 | 1.78E-01 | 0.235 |
| fast-mflux | 7.72E-02 | 1.47E-02 | 6.58E-04 | 1.78E+00 | 0.0455 |
| slow-sflux | 7.72E-03 | 1.08E-03 | 3.93E-05 | 4.16E-01 | 0.1225 |
| fast-sflux | 7.72E-02 | 1.08E-02 | 4.65E-05 | 4.16E+00 | 0.0304 |
| slow-ssflux | 7.72E-03 | 2.82E-03 | 3.08E-06 | 1.39E+00 | 0.0819 |
| fast-ssflux | 7.72E-02 | 2.82E-02 | 6.70E-07 | 1.39E+01 | 0.0226 |

| (b) Heterogeneous emissions | | | | | |
|---|---|---|---|---|---|
| Case | | | | | |
| dx = 1 km | | | | | |
| slow-1km-VV05 | 1.46E-04 | 1.46E-04 | 1.42E-03 | 1.42E-03 | 1.0184 |
| fast-1km-VV05 | 1.46E-03 | 1.46E-03 | 5.15E-03 | 5.15E-03 | 0.9094 |
| slow-1km-sflux | 3.53E-04 | 3.53E-04 | 1.41E-02 | 1.41E-02 | 0.5053 |
| fast-1km-sflux | 3.73E-03 | 3.73E-03 | 9.38E-02 | 9.41E-02 | 0.1365 |
| dx = 2 km | | | | | |
| slow-2km-VV05 | 1.21E-04 | 1.21E-04 | 1.52E-03 | 1.53E-03 | 0.9952 |
| fast-2km-VV05 | 1.22E-03 | 1.22E-03 | 5.95E-03 | 6.06E-03 | 0.7634 |
| slow-2km-sflux | 3.48E-04 | 3.48E-04 | 2.10E-02 | 2.13E-02 | 0.3068 |
| fast-2km-sflux | 3.46E-03 | 3.46E-03 | 1.70E-01 | 1.73E-01 | 0.0487 |
| dx = 6 km | | | | | |
| slow-6km-VV05 | 1.82E-04 | 1.82E-04 | 2.12E-03 | 2.14E-03 | 0.665 |
| fast-6km-VV05 | 1.80E-03 | 1.80E-03 | 1.37E-02 | 1.39E-02 | 0.2068 |
| slow-6km-sflux | 5.55E-04 | 5.55E-04 | 7.40E-02 | 7.52E-02 | 0.0312 |
| fast-6km-sflux | 5.40E-03 | 5.40E-03 | 7.13E-01 | 7.26E-01 | 0.0034 |

**Table 1:** The initial Kolmogorov Damköhler numbers $Da_{k\,A,i}$ and $Da_{k\,B,i}$, and final Kolmogorov Damköhler numbers $Da_{k\,A,f}$ and $Da_{k\,B,f}$ with the resultant normalised effective chemical reaction rate $k_{eff}/k$ in the DNS simulations with (a) homogeneous emissions and (b) heterogeneous emissions. Please refer to Table 2 of the manuscript for the corresponding Damköhler numbers.

[Figure]

**Figure 1:** Replot of Figure 8 in the manuscript. An additional $x$-axis with the limiting Kolmogorov Damköhler number ($Da_{k,lim}$) (top $x$-axis) and the results of the DNS simulations with NO-NO2-O3 chemical system (circles in green) are added. The latter will not be shown in the next revision of the manuscript.
* * *
**Review #5:**

b) At section 2.2.1, the authors focus on the most extreme case of segregation, i.e.an irreversible second-order reaction. I understand their choice, but I think the reader will appreciate if they could include in their chemical mechanism a backward reaction. By so doing, the chemical will become closer to an equilibrium, and it will probably reduce the effects of the segregation. In that respect, the analytical solution for three chemical species and two reactions derived by Jonker et al. (2004) (Characteristics length scales of reactive species in the convective boundary layer, Journal of Atmospheric Sciences 61, 41-55) can be very useful to study the effect without introducing additional computational costs. Note that variations on time and on space due to presence of clouds or vertical perturbances of the radiation transfer within the urban canopy can be interested cases to bring this triad of species out of equilibrium, and therefore enhance the segregation effect again.

**Response #5:**

(b) Section 2.2.1: We have performed another set of simulations with the NO-NO2-O3 chemical system with increasing NO emission fluxes, which include the backward reaction of NO2 → NO + O3. The details of the simulation settings can be found in Section 3.5 of the doctoral thesis of CWYL (available at
https://pure.mpg.de/pubman/faces/ViewItemOverviewPage.jsp?itemId=item_3069134). As a comparison we have plotted the resultant errors ($E$) in that set of simulations over the limiting Damköhler number ($Da_{lim}$) in Figure 1 in this document (green circles). One can see that the resultant errors are smaller in all cases with NO-NO2-O3 chemical system as compared to those in the A-B-C chemical system. However, as $Da_{lim}$ reaches $\sim O(10)$, the increase in $E$ is not as large as that with the A-B-C chemical system. The error increases to around 15%, instead of $> 80\%$ with the A-B-C chemical system. Also, this error does not increase with $Da_{lim}$ once the $Da_{lim}$ reached $O(10)$,

contrary to the increasing trend observed in the A-B-C chemical system. This phenomenon agrees with what was reported in Jonker et al. (2004). Unfortunately we cannot applied the analytical solution proposed in Jonker et al. (2004) to calculate the concentrations with a backward reaction, as in some of our simulations with large emission fluxes, the concentrations of Tracer A and B do not reach equilibrium at the end of the simulations (but the variances of the concentrations do). Therefore including a backward reaction may require reruns of the simulations. We therefore add the following highlighted text to discuss this above-mentioend summary points from the doctoral thesis of CWYL and add a suggestion to include a backward reaction in future simulation studies in the Discussion:

Unlike other studies (e. g. Ouwersloot et al. (2011); Dlugi et al. (2019); Kim et al. (2016); Li et al. (2016, 2017)) in which 30 multiple-reaction chemical systems are employed, our work mostly focuses on an idealised second-order chemical reaction of two non-specific reactive species. This approach allows us to interpret our work for any second-order chemical reactions. For a chemical species involved in a multiple-reaction chemical system, like $O_3$, one can still calculate the net impact of chemical segregation by considering the errors of all reactions in which the species is involved. However, it is also important to notice that the net impact of chemical segregation on such a species would in general be reduced with the increasing complexity of the chemical system, because cycling reactions tend to lengthen the chemical timescale and reduce the corresponding Damköhler number, and hence reduce the chemical segregation. For example, with emission fluxes of $NO_X$ comparable to our sflux and ssflux cases, the segregation coefficient between isoprene and OH is only -0.05 and -0.17 in the high- and very high-$NO_X$ cases of Kim et al. (2016) respectively. Li (2019) also performed simulations similar to this study with the $NO$–$NO_2$–$O_3$ triad, and found less significant errors than with the A-B-C chemical system. Also, this error does not increase with $Da_{lim}$ when $Da_{lim}$ reached the order of 10, contrary to the increasing trend observed in the A-B-C chemical system. Therefore we expect the magnitude of chemical segregation to be smaller than the values reported in this study when a multiple-reaction chemical system is included. Future studies can also consider implementing a backward reaction in addition to the idealised second-order chemical reaction to approximate the effect of a multiple-reaction chemical system.
* * *
**3. Results**

**Review #6:**

Homogeneous emission. For the sake of completeness, I think it might be interesting in Figure 2 to add an extra panel but now with a typical time average used by the air quality models (~30 minutes) to (probably) show how the small-scale fluctuations of the chemical term are also filtered out due to the time-averaging.

**Response #6:**

The time-averaged cross-section of the production term for the last 30 minutes of the simulation is shown in Figure 2 in this document. It will be included in the next revision of our manuscript.

[Figure]

[Figure]

**Figure 2:** The top panel of Figure 2 in the manuscript (*top panel*) and the color map of the same simulation with a time-averaged field of the production field over the last 30 minutes of the simulation (*bottom panel*).
* * *
**Review #7:**

Section 3.1.2. (lines 12-14). I think the authors are reporting a very interesting results regarding the more prominent minima due to the higher vertical resolution. Could they please elaborate and quantified a bit more?

**Response #7:**

Section 3.1.2: The maxima near the top of the surface layer are found at an altitude of around 60 m. While this height corresponds to around 8.73 vertical levels in our DNS model, it corresponds to 2.34 vertical levels in the LES model used in Vinuesa and Vilà-Guerau de Arellano (2005). As the segregation coefficient is relatively sensitive to the change in altitude in this region, the lower vertical resolution would underestimate the slope of the vertical profile of the segregation coefficient in that layer. Although the LES model includes subgrid parametrisation, it is anticipated to show similar effect as in the coarse-grid model analysis (compare the vertical profiles of $k_{eff}/k$ between the DNS model and the coarse-grid models in Figure 9).
* * *
**Review #8:**

Since the paper is entitled "errors", I think an additional figure 3 that can complete the discussion is the calculation of the exchange coefficient K defined as the ratio between the turbulent flux and the mean gradient). I realize that in the middle of the CBL can be undetermined, but close to the surface and at the entrainment zone it might provide information on the combined effect of chemistry and turbulence on the parameterization of the transport of chemical species.

**Response #8:**

Attached please see the plots of the exchange coefficients of Tracer A, B and C for the different cases addressed in Figure 3. For all Tracers, the magnitudes of their exchange coefficients are still more significant in the mixed layer. Their magnitudes at the surface and in the encroachment zone are still relatively small. At the surface, the no-slip condition was implemented in the DNS code, so that the turbulent fluxes are always equal to zero, resulting a zero exchange coefficient. In the encroachment zone, the large turbulent fluxes are counterbalanced by the large concentration gradient, so that the exchange coefficients are again relatively small. Note that the relatively "noisy" profiles of the exchange coefficients of Tracer B and C in the mixed layer are due to the very small concentration gradients of the two tracers in the mixed layer. The level of "noisiness" of these

profiles may hinder the meaningfulness of including these profiles in the discussion. However, the profiles of Tracer A show better "signal-to-noise" ratios. Also note that as the emission fluxes of Tracer A increases (mflux - ssflux), the corresponding exchange coefficients are almost the same between the slow- and fast-chemistry cases.

(Note that the following sentences are not shown in the original reply:)
Here we think that our results regarding the exchange coefficients are not well researched in our current work to be shown in the paper, so we propose to suggest this point as a possible future work in the Discussion section (See Response #11). We hope the reviewer and the editor can agree on this proposal.

[Figure]

Figure 3: The vertical profiles of the horizontally-averaged exchange coefficients ($\kappa$) of Tracer A (*left panel*), B (*middle panel*) and C (*right panel*) normalised by the product of the boundary layer height $z_i$ and the convective velocity $w_*$. Here the same simulation cases are shown as in Figure 3 of the manuscript. The data here are averaged every 8 vertical levels to lessen the effect of the noise.
* * *
**Review #9:**

Section 3.2 (line 29). It might be interesting to show how this Is = - 0.8 changes if a first-order backward reaction is included in the chemical mechanism of reaction R1.

**Response #9:**

Section 3.2 (line 29): Please refer to point 2(b) (Response #7) in this reply.
* * *
**Review #10:**

Figure 9. Could the authors elaborate more the reasons of lower values on k_eff/k at the entrainment zone compared to the values near the surface? Both regions are characterized by higher gradients of the chemically active species. Why are these differences at these two regions?

**Response #10:**

Figure 9: As the segregation can be related to the standard deviations of the concentrations of the tracers (Vinuesa and Vilà-Guerau de Arellano, 2005), we have attached the vertical profiles of the concentration variances of Tracer A, B and C in Figure 4 in this document. One can see that for all tracers in all cases, the concentration variances in the encroachment zone are always larger than those near the surface. This difference is even larger for Tracer B and C. Therefore one may anticipate a large segregation and hence a lower $k_{eff}/k$ in the encroachment zone than near the surface.

[Figure]

Figure 4: The vertical profiles of the concentration variances of Tracer A (*left panel*), B (*middle panel*) and C (*right panel*) normalised by the corresponding characteristic concentrations.

**4. Discussion**

**Review #11:**

15 I think it can be worth to make a discussion point that the errors are not only on the segregation and therefore on the chemical reaction rate, but also on the eddy diffusivity and therefore on the transport of chemical species

**Response #11:**
20 (Note that we have missed this response in the original reply.)

This would be an insightful point to add into the Discussion. We have added accordingly the highlighted text in the corresponding paragraph in Section 5.1:

25 One important aim of the study of chemistry-turbulence interaction is to provide a correction to the error in large-scale model induced by neglecting subgrid chemical segregation. While our work suggests a correction of $k_{eff}$ with dependencies on $Da_{lim}$ and dx, other work suggest that such correction should also include other variables such as updraft/downdraft fluxes (Petersen and Holtslag, 1999), variance of the reacting species, entrainment/emission fluxes (Petersen and Holtslag, 1999; Vinuesa and Vilà-Guerau de Arellano, 2003), turbulent fluxes (Dlugi et al.,
30 2014), variance of the emission (Galmarini et al., 1997), magnitude and direction of mean horizontal wind (Ouwersloot et al., 2011) and distance from the sources (Karamchandani et al., 2000). The correction to the vertical exchange coefficient (i.e. the ratio between the turbulent flux and the mean gradient) can also serve as an alternative way to address the change in the transport terms from the effect of chemical reactions (Vilà-Guerau de Arellano and Duynkerke, 1992; Geyer and Stutz (2004)). There have been attempts to implement a parametrisation of subgrid
35 chemistry-turbulence interaction to large-scale models. For example, Molemaker and Vilà-Guerau de Arellano (1998) suggested the use of lookup tables to indicate how chemical reaction rates should be corrected under different physical or chemical scenarios. Lenschow et al. (2016) developed a one-dimension second-order closure model to account for the vertical turbulent mixing of chemical species ready to be incorporated into large-scale models. Given that the effect of subgrid chemical segregation is non-negligible under urban conditions, modellers
40 should consider applying similar parametrisations in areas with intense emission and large source heterogeneity.

**Review #12:**

Section 5.1. (lines 9-16) Will look-up tables depending on the processes that they have mentioned be a solution instead of the parameterizations?

**Response #12:**

Section 5.1 : As the chemical segregation is affected by many initial and boundary conditions and also the combination of these conditions, it is very difficult to formulate a parametrisation that can be applied to every scenario. Under these circumstances, look-up tables that describe the segregation or $k_{eff}/k$ under specific conditions may serve as an alternative to parametrisations so that modellers can choose the right tables that best describe the scenarios they want to simulate. The following highlighted text are added in the corresponding paragraph in Section 5.1:

One important aim of the study of chemistry-turbulence interaction is to provide a correction to the error in large-scale model induced by neglecting subgrid chemical segregation. While our work suggests a correction of $k_{eff}$ with dependencies on $Da_{lim}$ and dx, other work suggest that such correction should also include other variables such as updraft/downdraft fluxes (Petersen and Holtslag, 1999), variance of the reacting species, entrainment/emission fluxes (Petersen and Holtslag, 1999; Vinuesa and Vilà-Guerau de Arellano, 2003), turbulent fluxes (Dlugi et al., 2014), variance of the emission (Galmarini et al., 1997), magnitude and direction of mean horizontal wind (Ouwersloot et al., 2011) and distance from the sources (Karamchandani et al., 2000). The correction to the vertical exchange coefficient (i.e. the ratio between the turbulent flux and the mean gradient) can also serve as an alternative way to address the change in the transport terms from the effect of chemical reactions (Vilà-Guerau de Arellano and Duynkerke, 1992; Geyer and Stutz (2004)). There have been attempts to implement a parametrisation of subgrid chemistry-turbulence interaction to large-scale models. For example, Molemaker and Vilà-Guerau de Arellano (1998) suggested the use of lookup tables to indicate how chemical reaction rates should be corrected under different physical or chemical scenarios. Lenschow et al. (2016) developed a one-dimension second-order closure model to account for the vertical turbulent mixing of chemical species ready to be incorporated into large-scale models. Given that the effect of subgrid chemical segregation is non-negligible under urban conditions, modellers should consider applying similar parametrisations in areas with intense emission and large source heterogeneity.
* * *
**Review #13:**

Page 17 and line 1. Will previous work on natural canopies be useful to be connectedto urban canopies?

**Response #13:**

P. 17 Line 1: Previous work on natural canopies will definitely be useful to be connected to urban canopies. First of all, we can use the results on natural canopies to estimate/extrapolate the effect on urban canopies. Second, as the vegetation are always mixed within the urban canopies in many cities, it would be also useful to combine the features of natural and urban canopies in future simulations on the topic. The following highlighted text are added in Section 5.2:

An important source of surface forcings in an urban boundary layer is undoubtedly from the urban structures (buildings and streets in the urban canopy). The structure of turbulent flow can be significantly altered in the street canyons due to the perturbation in radiation and the exchange of heat and momentum in the urban structures (e. g. Oke (1997)). These urban canopy meteorological forcings also affect vertical turbulent transport and hence on the vertical distribution of chemical species (Huszar et al., 2020). Therefore, our DNS simulations are more suitable in addressing the vertical mixing caused by the turbulent motions in the mixed layer relatively far away from the surface features that may induce other additional forcings. But in the mixed layer, the urban canopy still potentially affects the chemistry and dynamics in the boundary layer by means of surface roughness and emission heterogeneity. While in this work we have addressed the effect of emission heterogeneity, the effect of surface roughness and other configurations of emission patterns (e. g. Auger and Legras (2007)) can be further studied in the future. Previous work on natural canopies can also be combined to further studies with urban canopies as the vegetation are mixed within the urban canopies in many cities.

**Review #14:**

Since section 5.2 presents an outlook of possibilities for future research, I think it might important to include a short description on the effects of radiation and its perturbation(clouds, urban canopy) closely couple to the turbulent effect. An additional experiment including the backward reaction as here proposed can pave the way to future research.

**Response #14:**
(Note that we have missed this response in the original reply.)

We have mentioned to consider the effects of clouds and urban canopy in the future work in separate points in Section 5.2, but we did not classified these effects as the effects of radiation and also did not explicitly point out that these perturbations are closely coupled to the turbulent effect. We therefore add the highlighted sentence to illustrate this point with the existence of clouds in the first paragraph of Section 5.2 to address these points:

Due to computational limitations of the DNS simulations, some other conditions that may be important for turbulent mixing in the urban boundary layer were neglected in our work. First of all, the growth of the boundary layer is driven by a constant buoyancy flux, so that the boundary layer height gradually increases. Hence, the simulation can only approximate the time when the boundary layer grows from sunrise to mid-afternoon. This also raises the issue of the time required for statistical equilibrium to attain. In some cases, this time may be longer than the duration of daylight, after which the atmosphere is no longer convective. This poses a greater problem to cases with strong emission fluxes, as the time required to attain statistical equilibrium is even longer. It may not be practical to run longer than the duration of daylight even though statistical equilibrium is not reached. Second, our simulations only address a convective boundary layer under clear-sky conditions. We do not take other scenarios with different weather conditions (such as cloud-top boundary layer, e. g. Li et al. (2016, 2017)) or when the atmosphere is stably stratified (typical for nighttime, e. g. Galmarini et al. (1997); Geyer and Stutz (2004)) into account. For instance, the perturbation in radiation with the existence of clouds closely couples to the turbulent effect (Mellado, 2017) and also modifies chemical reactions (Barth et al. (2002); Vilà-Guerau de Arellano et al. (2005)). Third, it may be useful to consider other boundary conditions of the entrained tracers, such as varied entrainment fluxes (e. g. Krol et al. (2000); Albrecht et al. (2016)) and varied concentrations in the free troposphere due to long-range transport (e. g. Zyryanov et al. (2012)). Fourth, our simulations also do not include any mean horizontal winds (e. g. Ouwersloot et al. (2011)), or forcings from the surface other than the constant buoyancy flux, such as varied surface heat fluxes (e. g. Ouwersloot et al. (2011); Van Heerwaarden et al. (2014)) and surface roughness.

The perturbation in radiation within the urban canopy is addressed in the following highlighted text:

An important source of surface forcings in an urban boundary layer is undoubtedly from the urban structures (buildings and streets in the urban canopy). The structure of turbulent flow can be significantly altered in the street canyons due to the perturbation in radiation and the exchange of heat and momentum in the urban structures (e. g. Oke (1997)). These urban canopy meteorological forcings also affect vertical turbulent transport and hence on the vertical distribution of chemical species (Huszar et al., 2020). Therefore, our DNS simulations are more suitable in addressing the vertical mixing caused by the turbulent motions in the mixed layer relatively far away from the surface features that may induce other additional forcings. But in the mixed layer, the urban canopy still potentially affects the chemistry and dynamics in the boundary layer by means of surface roughness and emission heterogeneity. While in this work we have addressed the effect of emission heterogeneity, the effect of surface roughness and other configurations of emission patterns (e. g. Auger and Legras (2007)) can be further studied in the future. Previous work on natural canopies can also be combined to further studies with urban canopies as the vegetation are mixed within the urban canopies in many cities.

Please refer to Response #5 for the last point on backward reaction.
* * *

30   the $x$-axis to indicate that in the next revision of the manuscript as in the following figure:

[Figure]
* * *
**Review #3:**

5    Figure 4&5. Is there a particular reason you use Is in Fig. 5 and keff/k in Fig. 4? Could you keep it consistent?

**Response #3:**

10    The segregation coefficient $I_S$ is presented in Figure 5 to match with the description in Section 3.2. We think that it will be easier to explain that the flows of Tracer A and B are co-related in the situation with $I_S > 0$ with the expression of $I_S$ instead of $k_{eff}$. We suggest to keep the $x$-axis in $I_S$ but to add an additional $x$-axis in $k_{eff}$ in Figure 5 for consistency as the below figure:

[Figure]
* * *
**Review #4:**

Figure 9. I wonder if you run the simulations with the fast chemistry. If so, how different are they from the slow chemistry cases shown here?

**Response #4:**

We have also performed similar runs for the fast chemistry cases in Figure 9. With fast chemistry, the underestimation of $[E]$ would be more significant, but the locations with the most prominent
25    underestimation are still at the top of the surface layer and at the top of the boundary layer. We will present the fast chemistry case with the mflux emission flux scenario in Figure 9 for a comparison in the next revision of the manuscript as in the following figure:

[Figure]

**Review #5:**

5   Equation1. Please define $F_b$.

**Resposne #5:**

$F_b$ is in fact defined on Page 5 Line 8.

**Review #6:**

Page 8 Line 12. Add the acronym CBL to where the full name first appears.

**Response #6:**

Thanks for the reminder. We will add the definition of the acronym CBL when the term "convective boundary layer" first appears on Page 2 Line 21:

> Many of these LES studies focus on the convective boundary layer ==(CBL),== in which the imbalance between updraft and downdraft transport produces a large segregation of the reactants (Wyngaard and Brost, 1984; Chatfield and A. Brost, 1987).

**Review #7:**

Page 13 Line 4. Can you clarify "interpolated"?

30   **Response #7:**

The high-resolution grids of the DNS simulations are mapped onto the lower-resolution grids of the coarse-grid models. The concentration fields obtained from the DNS simulations are then interpolated from the high-resolution DNS model grids to the lower-resolution coarse-grid model
35   grids using local-area averaging. We have rephrased the description of the coarse-grid model method in Page 13 Line 14-15:

[revised manuscript text omitted]